# Genome-wide CRISPR off-target prediction and optimization using RNA-DNA interaction fingerprints

Qinchang Chen [1,2,3,8], Guohui Chuai[2,4,8], Haihang Zhang [5,8], Jin Tang[3], Liwen Duan[3], Huan Guan[3], Wenhui Li [3], Wannian Li[1], Jiaying Wen[1], Erwei Zuo [5] ✉, Qing Zhang[6,7] ✉ & Qi Liu [1,2,3,4] ✉

The powerful CRISPR genome editing system is hindered by its off-target effects, and existing computational tools achieved limited performance in genome-wide off-target prediction due to the lack of deep understanding of the CRISPR molecular mechanism. In this study, we propose to incorporate molecular dynamics (MD) simulations in the computational analysis of CRISPR system, and present CRISOT, an integrated tool suite containing four related modules, i.e., CRISOT-FP, CRISOT-Score, CRISOT-Spec, CRISORT-Opti for RNA-DNA molecular interaction fingerprint generation, genome-wide CRISPR off-target prediction, sgRNA specificity evaluation and sgRNA optimization of Cas9 system respectively. Our comprehensive computational and experimental tests reveal that CRISOT outperforms existing tools with extensive in silico validations and proof-of-concept experimental validations. In addition, CRISOT shows potential in accurately predicting off-target effects of the base editors and prime editors, indicating that the derived RNA-DNA molecular interaction fingerprint captures the underlying mechanisms of RNA-DNA interaction among distinct CRISPR systems. Collectively, CRISOT provides an efficient and generalizable framework for genome-wide CRISPR off-target prediction, evaluation and sgRNA optimization for improved targeting specificity in CRISPR genome editing.

The CRISPR genome-editing system, for example the CRISPR-Cas9 system, has broad applications in biology and medicine[1,2]. Although such an RNA-guided system permits precise genome editing, it presents a great challenge for off-target effects[3]. Studies have demonstrated that Cas9 can tolerate a number of base pair (bp) mismatches, and incorrectly cleave sites do not fully complement the sgRNA (off-target sites)[4]. Off-target effects can lead to potential side effects, which will hinder the development and clinical applications of CRISPR system.

[1]Key Laboratory of Spine and Spinal Cord Injury Repair and Regeneration (Tongji University), Ministry of Education, Orthopaedic Department of Tongji Hospital, Frontier Science Center for Stem Cell Research, Bioinformatics Department, School of Life Sciences and Technology, Tongji University, Shanghai 200092, China. [2]Translational Medical Center for Stem Cell Therapy and Institute for Regenerative Medicine, Shanghai East Hospital, Frontier Science Center for Stem Cell Research, Bioinformatics Department, School of Life Sciences and Technology, Tongji University, Shanghai 200092, China. [3]Research Institute of Intelligent Computing, Zhejiang Lab, Hangzhou 311121, China. [4]Shanghai Research Institute for Intelligent Autonomous Systems, Shanghai 201210, China. [5]Shenzhen Branch, Guangdong Laboratory of Lingnan Modern Agriculture, Key Laboratory of Gene Editing Technologies (Hainan), Ministry of Agriculture and Rural Affairs, Agricultural Genomics Institute at Shenzhen, Chinese Academy of Agricultural Sciences, Shenzhen, China. [6] Roche R&D Center (China) Ltd., China Innovation Center of Roche, Shanghai 201203, China. [7]Present address: Ailomics Therapeutics, Shanghai 201203, China. [8]These authors contributed equally: Qinchang Chen, Guohui Chuai, Haihang Zhang. ✉e-mail: zuoerwei@caas.cn; qing.zhang@ailomics.com; qiliu@tongji.edu.cn

Many in silico tools have been developed to design sgRNAs to avoid off-target effects. Generally, the related methods can be divided into two categories: hypothesis-driven (*e.g.* CRISPRoff[5], uCRISPR[6], MIT[7] and CFD[8]) and learning-based (e.g., deepCRISPR[9], CRISPRnet[10] and DL-CRISPR[11]). Hypothesis-driven methods score off-target effects using empirically derived rules, while learning-based methods predict off-target effects by using machine learning models. Despite their differences, these tools achieved limited performance in genome-wide off-target prediction due to the lack of deep understanding and investigating of the CRISPR molecular mechanism. The molecular mechanism of Cas9 mainly includes two essential processes:[12–15] 1) the binding of a Cas9-sgRNA riboprotein to the target sequence to form a stable complex of Cas9-sgRNA-DNA; and 2) the allostery of the two nuclease domains of Cas9 to cleave the DNA. These two processes are driven by a series of intermolecular and intramolecular interactions within the Cas9-sgRNA-DNA complex, including the protein-protein[16], nucleic acid-protein[17] and nucleic acid-nucleic acid[18] molecular interactions. Such molecular interactions can be used to predict CRISPR editing processes as well; however, limited works have been presented. It has been reported that the binding energy of the Cas9-sgRNA-DNA complex, which is a kind of molecular interaction derived from experimental measurements, has enabled more accurate CRISPR off-target predictions[5,6], while such molecular interactions are waiting to be further elucidated for improved genome-wide CRISPR off-targeting prediction.

To characterize the RNA-DNA molecular interaction features of the CRISPR system and address the issue of accurate prediction of the genome-wide CRISPR off-target effects, in this study, we propose CRISOT (<u>CRIS</u>PR <u>O</u>ff-<u>T</u>arget), an integrated computational framework for genome-wide CRISPR off-target prediction, evaluation and optimization based on the RNA-DNA molecular interactions of the CRISPR system. The main contribution of this study is to derive an efficient and generalizable RNA-DNA molecular interaction fingerprint characterizing the underlying interaction mechanisms of the RNA-DNA hybrid of CRISPR system by molecular dynamics (MD) simulations[19], which is a mature computational method for investigating the molecular mechanism and the molecular interactions of biomolecules[20]. The idea of using MD simulations and molecular interactions to empower CRISPR technology was recently highlighted[19,21]. Although several studies existed to performed MD simulations on the Cas9 system[13,22–24], its full potential in CRISPR off-target modeling has not been evacuated. In addition, the previous off-target reducing strategies, e.g., protein engineering[25–27] and delivering additional sgRNAs[28], are complex and require additional experimental works, indicating a requirement for a simple yet powerful strategy to improve targeting specificity. Furthermore, by incorporating MD simulations as a prior in the CRISPR modeling, the issue of lack of sufficient off-target labeling information in the building of off-target prediction model can be alleviated. To this end, CRISOT contains four related modules, i.e., CRISOT-FP, CRISOT-Score, CRISOT-Spec, CRISOT-Opti for RNA-DNA molecular interaction fingerprint generation, genome-wide CRISPR off-target prediction, sgRNA specificity evaluation and sgRNA optimization of Cas9 system respectively. Our comprehensive computational and experimental tests revealed that CRISOT exhibited great advances over existing tools. In addition, CRISOT showed potential in accurately predicting off-target effects of the base editors and prime editors, indicating that the derived RNA-DNA molecular interaction fingerprint captures the underlying mechanisms of RNA-DNA interaction among distinct CRISPR systems. Collectively, CRISOT provides an efficient and generalizable system for genome-wide CRISPR off-target prediction, evaluation and sgRNA optimization for improved targeting specificity in CRISPR genome editing.

## Results

### Conceptual framework of CRISOT

The conceptual framework of CRISOT is shown in Fig. 1 and the related methods are described in the Methods section. CRISOT is initially designed for CRISPR-Cas9 system which involves the interaction of RNA-DNA hybrids, while its potential utility in other RNA-guided CRISPR system including base editors and prime editors are also demonstrated. Since the interaction of RNA-DNA hybrid is the key for Cas9 activation[12,29–31](Fig. 1a), the basic idea of CRISOT is to understand the molecular interactions of the RNA-DNA hybrid at atom level using MD simulations, and to incorporate such information in the genome-wide CRISPR off-target prediction, evaluation and optimization.

Specifically, we firstly calculated several kinds of molecular interaction features from the MD trajectories of RNA-DNA hybrids, including hydrogen bonding, binding free energies, atom positions (atom-atom distances, angles and dihedral angles) and base pair/base step geometric features (Fig. 1b). These features were used to derive the interaction fingerprints (CRISOT-FP) of an RNA-DNA hybrid (Fig. 1c). We collected annotated off-target datasets from various genome-wide off-target sequencing including Change-seq, Site-seq, Circle-seq, Guide-seq etc. and trained XGBoost (XGB) classification models using CRISOT-FP, which is proven to outperform existing off-target predicting methods in the following study. Then, three related flexible modules based on CRISOT-FP were developed, including (1) CRISOT-Score was developed by identifying key features derived from CRISOT-FP models, which can quickly calculate the off-target score of a given pair of sgRNA and off-target sequences (Fig. 1d). (2) CRISOT-Spec was developed to calculate the specificity score of a given sgRNA (Fig. 1e), by aggregating the CRISOT-Scores of the high-scored off-target sequences among all possible off-target sites. (3) CRISOT-Opti was developed for sgRNA optimization (Fig. 1f). For an sgRNA with high editing efficiency and poor targeting specificity, CRISOT-Opti introduces a single nucleotide mutation for this given sgRNA by reducing the off-target effect of the given sgRNA while maintaining its on-target effect.

### CRISOT-FP serves as a generalizable molecular interaction fingerprint to represent RNA-DNA hybrids in Cas9 system

**Designing of CRISOT-FP.** We proposed that the RNA-DNA hybrids contain information on molecular interactions governing the activation of Cas9. Therefore, we systematically designed a series of RNA-DNA hybrids and performed MD simulations to develop the RNA-DNA molecular interaction features (Fig. 2a). Detailed descriptions on the development and engineering of RNA-DNA molecular interaction features can be found in Methods section (Fig. 2b, c and Figs. S1, 2). As a result, a total of 193 features (Supplementary Data 1) were collected to encode sgRNA-DNA hybrids (Fig. 2c). We encoded each pair of sgRNA and DNA nucleotides with the 193 molecular interaction features (Supplementary Data 1). The 20-bp sgRNA-DNA hybrid was therefore encoded with 193*20 = 3860 features, which made up the RNA-DNA molecular interaction fingerprints, i.e., CRISOT-FP, of the hybrid. Because the sequences were position-dependent, the encoding process made the CRISOT-FP position-dependent. We proposed that CRISOT-FP could be applied in various machine learning models as a basic and generalizable feature representation to capture the essential of RNA-DNA hybrids in Cas9 system. This point is demonstrated in the subsequent analysis.

**Comparison of CRISOT-FP with state-of-the-art feature encoding methods.** To evaluate the performance of CRISOT-FP, we curated the Group I benchmark datasets derived from in vitro genome-wide off-target detection techniques including Change-seq and Site-seq datasets (Table 1). The sgRNAs in these datasets are different from each other, thus they are independent to each other. Detailed description on the Group I datasets was given in Methods.

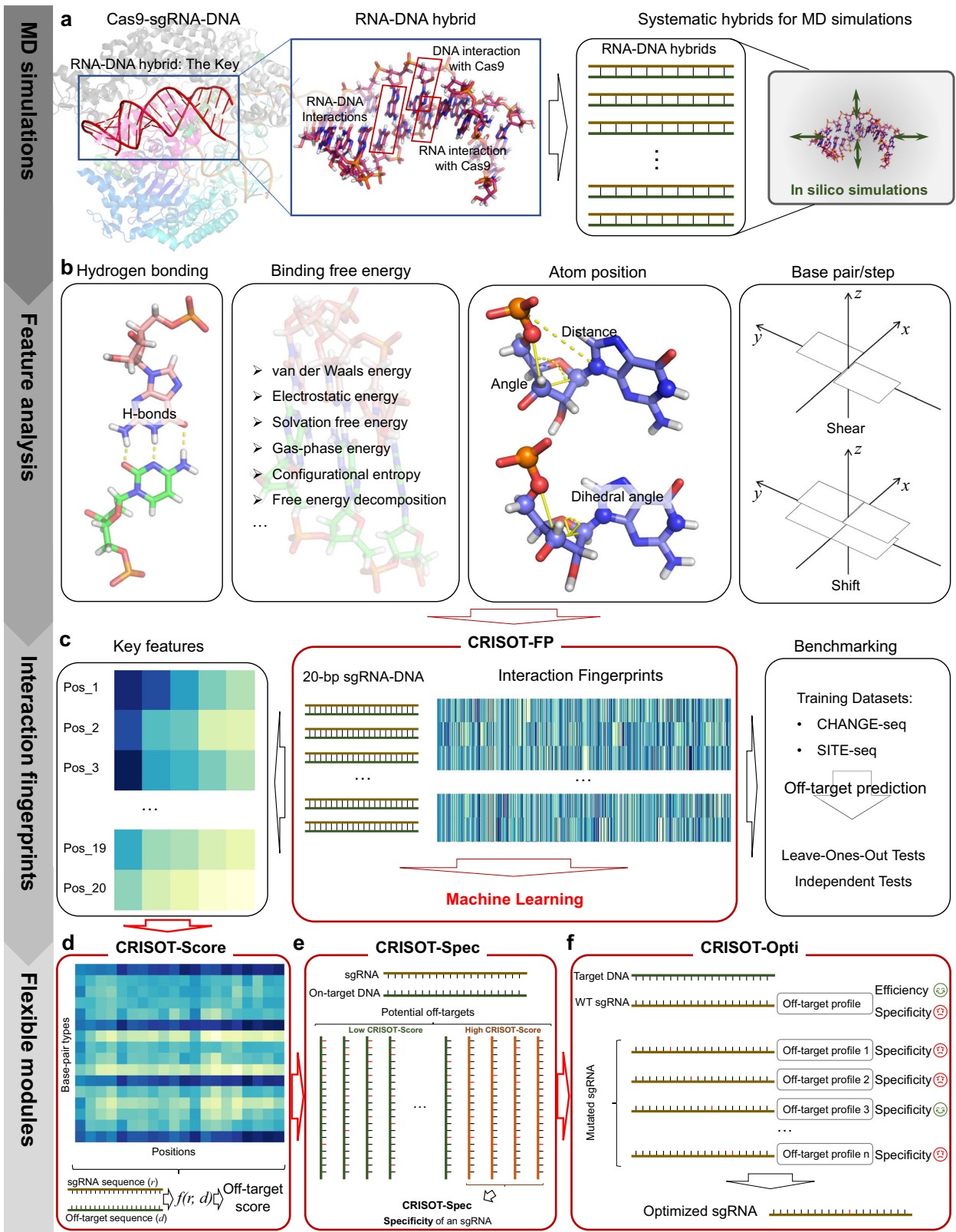

**Fig. 1 | Overview of the conceptual framework of CRISOT. a** Left, structure of the Cas9-sgRNA-DNA complex. The sgRNA-DNA hybrid is colored red and the interactions are detailed in the box. Right, conceptual representation of the systematic MD simulations. **b** Illustration of the interaction features. From left to right: hydrogen bonding, binding free energy, atom position, base pair/step geometric features. **c** Middle panel, encoding of RNA-DNA molecular interaction features to CRISOT-FP for machine learning. Right panel, benchmarking the CRISOT-FP models on different benchmark datasets. Left panel, analysis of the key RNA-DNA molecular interaction features. **d** Key features are used to develop CRISOT-Score for off-target scoring. **e** For a given sgRNA, the genome-wide off-target sites are scored by CRISOT-Score, and those high-scored off-target sites are summarized to derive a specificity evaluation schema - CRISOT-Spec. CRISOT-Spec can be used to evaluate the targeting specificity and the quality of the sgRNA. **f** When the specificity of an sgRNA is not satisfactory, CRISOT-Opti introduces single nucleotide mutations to change the off-target profile. By comparing different mutated sgRNAs, CRISOT-Opti can determine the optimized sgRNA as a substitution for the wild-type (WT) sgRNA.

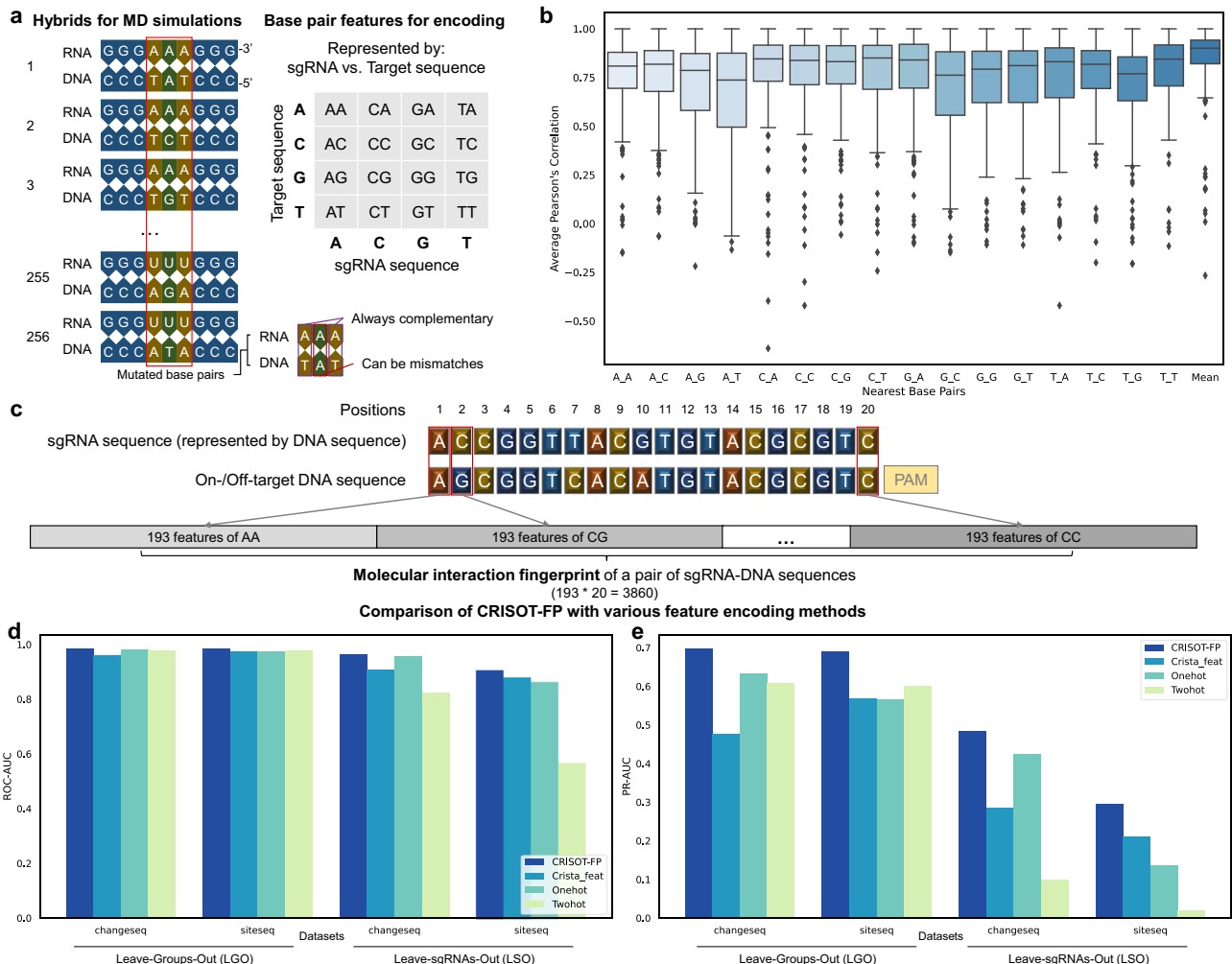

**Fig. 2 | Development and evaluation of CRISOT-FP. a** Conceptual representation of the RNA-DNA hybrids for MD simulations and feature analysis. Among the 9 bps of the 256 hybrids, only the 3 bps in the middle were mutated differently, generating match or mismatch bps for the middle bp and keeping the other two bps complementary. The features of the middle bps are calculated. The mean values are calculated for the 16 bp types. **b** Box plots of the average Pearson's correlations of all features. They indicate the capability of a series of features to represent those features with different nearest bps. Center bar, median; box, inter-quartile range; whiskers, low and high quartiles; $n = 246$ features (see Methods). **c** Encoding of an sgRNA-DNA hybrid to form the RNA-DNA molecular interaction fingerprints (CRISOT-FP). Every bp is encoded by the corresponding RNA-DNA molecular interaction features, so a 20-bp sgRNA-DNA hybrid is encoded by a 3860-bit fingerprints. **d**, **e** Comparison of CRISOT-FP with Crista-feat, one-hot and two-hot encoding methods by LGO and LSO evaluations on the Group I (Change-seq and Site-seq) datasets. ROC-AUC (**d**) and PR-AUC (**e**) results of LGO and LSO are presented. Source data are provided as a Source Data file.

We compared the performance of CRISOT-FP in CRISPR off-target prediction with three state-of-the-art feature encoding schemes, i.e., Crista_feat, One-hot and Two-hot encoding. Crista_feat features are the features used by CRISTA[32], which contain a wide range of features, including genomic content features, thermodynamics of the sgRNA, and the pairwise sgRNA-target similarity et al. One-hot and Two-hot encoding methods converts every bp of the sgRNA-target hybrid into a 16-bit and 8-bit vector (Fig. S2). We performed two leave-ones-out tests to evaluate CRISOT-FP and the existing encoding methods (Fig. S3). The leave-group-out (LGO) test randomly held out 1/5 of the inputs as testing data, so that the training and testing datasets contain all sgRNAs and different sequences of their corresponding on-/off-target sites. The leave-group-out (LGO) test randomly held out 1/5 of the sgRNAs and the corresponding on-/off-target sequences as testing data, so that training and testing datasets contained different sgRNAs and off-target sequences. In this case, LSO is taken as a stricter and more challenging prediction task compared to LGO. The LGO test estimated the prediction on unseen off-target sequences, whereas the

LSO test ensured that the sgRNAs and off-targets were totally different between training and testing datasets.

We applied the XGB[33] classification algorithm for both tests (Fig. 2d, e and Fig. S4) for its great prediction performance and model interpretation ability (see Methods). For the LGO tests, both CRISOT-FP and the sequence-based encodings showed good performance in terms of area under curve (AUC) analysis. CRISOT-FP achieved slightly better receiver operating curve (ROC) performance than the previous features, with ROC-AUCs higher than 0.984 (Fig. 2d). The precision-recall (PR) analysis indicated that CRISOT-FP achieved remarkably higher PR-AUCs than the other encoding methods, with PR-AUC scores of 0.698 and 0.690 for Change-seq and Site-seq datasets, respectively (Fig. 2e). For the LSO tests, CRISOT-FP still performed better than the previous features, and the differences in both ROC and PR were more remarkable than those in the LGO tests (Fig. 2d, e). Furthermore, we also tested CRISOT-FP and the other feature encoding methods by using different machine learning algorithms (Fig. S5). In this case, CRISOT-FP trained within the XGB algorithm outperformed the other features and the other machine learning algorithms. These results

**Table 1 | Details of the benchmark datasets**

| Group | Dataset | Experiment | Subset | sgRNAs | Genome-wide Potential Off-targets | Detected Off-targets | Ref. |
|---|---|---|---|---|---|---|---|
| I | Change-seq | in vitro | - | 110 | 1384929 | 21645 | 68 |
|  | Site-seq | in vitro | - | 12 | 153478 | 2047 | 14 |
| II | Circle-seq | in vitro | K562 | 6 | 60658 | 751 | 60 |
|  |  |  | HEK293 | 4 | 57026 | 857 |  |
|  |  |  | U2OS | 5 | 92113 | 2844 |  |
|  | Guide-seq | in cell | Tsai | 10 | 158169 | 341 | 37 |
|  |  |  | Listgarten | 23 | 187934 | 53 | 38 |
|  | Surro-seq | Targeted in cell | - | 105 | 6709 | 819 | 39 |
|  | TTISS | in cell | - | 59 | 669648 | 866 | 40 |
| III | PE2 | in vitro | - | 9 | 120021 | 1493 | 55 |
|  | BE3 | in vitro | Kim | 7 | 93510 | 76 | 58 |
|  |  | in vitro | Liang | 2 | 19917 | 3 | 57 |
|  | ABE7.10 | in vitro | Kim | 7 | 63306 | 212 | 56 |
|  |  | in vitro | Liang | 6 | 94902 | 44 | 57 |

indicate the great potential of CRISOT-FP to identify real off-targets among the background sequences in a highly imbalanced scenario. Therefore, the XGB models trained on the Group I datasets in this section were used for further evaluations on the independent datasets.

**Comparison of CRISOT-FP with state-of-the-art off-target prediction methods.** We further evaluated the ability of CRISOT-FP to predict genome-wide off-target effects in independent testing datasets (the Group II datasets, Table 1, see Methods) with no overlaps with the training datasets, and compared with state-of-the-art off-target prediction methods. The score values of CRISOT-FP were the means of the leave-ones-out models trained on Group I datasets described in the first benchmark study. Four hypothesis-driven off-target scoring tools (KinPred[34], CRISPRoff[5], uCRISPR[6] and CFD[8]) and four learning-based off-target prediction tools (MOFF[35], CRISPRnet[10], DLcrispr[11] and CNN_std[36]) were used for comparison. The learning-based models were taken from their original studies that were trained on their own datasets. Two of the compared learning-based methods, i.e., DLcrispr and CRISPRnet models, were trained on datasets containing several sgRNAs that were included in the testing datasets. For fair comparisons, we presented both the results on the complete testing datasets and the results excluding the overlapped sgRNAs for these models (Fig. 3 and Fig. S6–S11). In the case that all testing sgRNAs were contained in the training datasets of the compared methods, we have indicated this case by using the bars in gray. In the case that the testing sgRNAs partially overlapped with the training datasets of the compared methods, we have presented additional bars in slash hatch. The overlapped sgRNAs are summarized in Supplementary Data 3. Since the off-target datasets were highly imbalanced, the PR-AUC results rather than ROC-AUC were more convincing. Three benchmarking scenarios were performed.

In the first evaluation scenario, we tested the performance of CRISOT-FP in predicting the in vitro Circle-seq datasets (Fig. 3a, Figs. S6, S7). Although the performances of DLcrispr and CRISPRnet (the gray bars in Fig. 3a and Fig. S7) were higher/comparable to those of CRISOT-FP, this is attributed to that the testing sgRNAs were already contained in their training datasets, thus the colors of their bars were changed to gray to indicate potential data leakage. The MOFF model achieved the best performance among the other compared methods. The PR-AUC results showed that the MOFF model achieve comparable performance on circleseq_u2os dataset, while CRISOT-FP surpassed it in the other three datasets. The fair comparison of CRISOT-FP with the MOFF and CNN_std models and the hypothesis-driven methods showed that CRISOT-FP outperformed them, with an average PR-AUC of 0.46 ($p < 0.05$, Fig. S7c). Therefore, the CRISOT-FP achieved better

performance than the compared methods in predicting the in vitro off-target datasets.

In the second evaluation scenario, we used the in cell datasets (Table 1, see Methods) for evaluation, which include the Guide-seq[37,38], Surro-seq[39] and TTISS[40] datasets that are independent to the training datasets. The results (Fig. 3b, Figs. S8, S9) showed that CRISOT-FP outperformed all learning-based models and hypothesis-driven methods on the PR-AUC results ($p < 0.05$, Fig. S9c), even though data leakage may exist because the training datasets of the DLcrispr and CRISPRnet models contained sgRNAs that were in the testing Guide-seq datasets (Supplementary Data 3). For a fair comparison, we removed the sgRNAs in the testing datasets that were contained in the training datasets of DLcrispr and CRISPRnet, and the same sgRNAs were removed in the testing datasets for CRISOT-FP. The results showed that excluding the overlapped sgRNAs slightly decreased the performances of DLcrispr and CRISPRnet while increased those of CRISOT-FP (bars with slash hatch in Fig. 3b and Fig. S9). Therefore, the CRISOT-FP achieved better performance than the compared methods in predicting the in cell off-target datasets.

In the third evaluation scenario, we set 6 fold-change (FC) cutoffs to the targeted dataset (the Surro-seq[39] dataset) to obtain 6 datasets with various sensitivities (see Methods). The results (Fig. 3c, Figs. S10, S11) showed that the performances of all methods but CRISPRoff slightly increased as the FC cutoff increased. CRISOT-FP surpassed all learning-based models and hypothesis-driven methods on the PR-AUC results ($p < 0.05$, Fig. S11c). The results indicated that CRISOT-FP achieved better performance than all compared methods in predicting the targeted Surro-seq datasets with various sensitivities.

Taking together, the benchmark validation results indicated the high generalization capability of CRISOT-FP, and that CRISOT-FP outperformed the existing learning-based and hypothesis-driven methods in predicting the off-target effects of Cas9.

## CRISOT-Score provides effective off-target scoring of Cas9 system

**Feature importance analysis and the designing of CRISOT-Score.** We started our study by identifying the key features related to CRISPR off-target scoring. To this end, the XGB models were connected with SHapley Additive explanation (SHAP) algorithm[41] to identify the key interactions among all the features used in CRISOT-FP (see Methods). A higher SHAP importance value indicates a higher contribution of the feature to the predicted result. Some of the key features showed high SHAP importance values at most bp positions, indicating that these features and the related RNA-DNA molecular interactions were shared by different positions of the RNA-DNA hybrid in Cas9 (Fig. 4a,

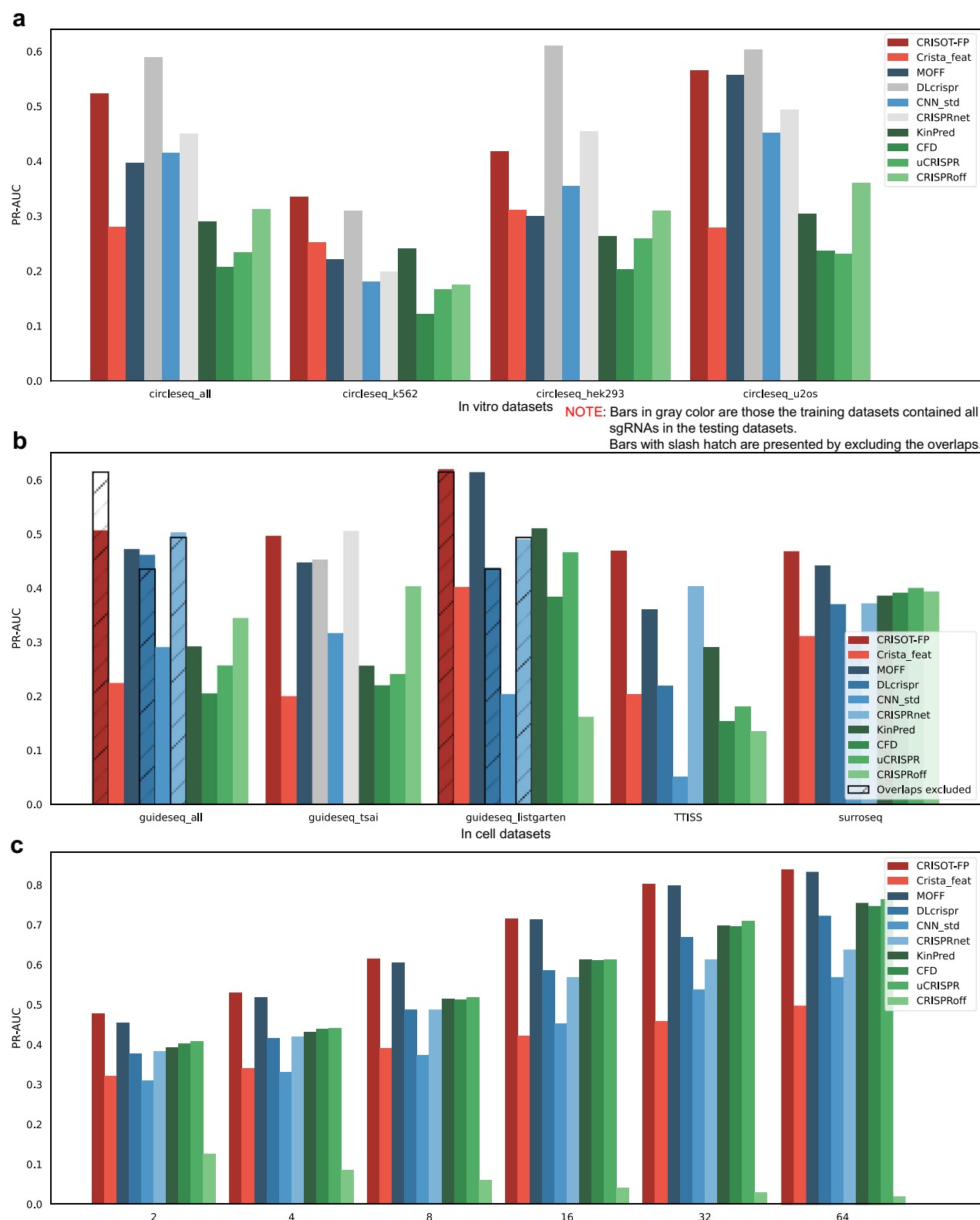

**Fig. 3 | Comparison of CRISOT-FP with the state-of-the-art off-target prediction methods in predicting independent (the Group II) datasets.** PR-AUC results for (**a**) in vitro datasets, (**b**) in cell datasets and (**c**) targeted (Surro-seq) dataset with different cutoff of fold-change (FC) are shown, and the ROC-AUC results can be found in Fig. S6–11. The blue and green bars represent learning-based and hypothesis-driven methods, respectively. Bars in gray color are those the training datasets contained all sgRNAs in the testing datasets. Bars with slash hatch are presented by excluding the overlapped sgRNAs. Of note, for fair comparisons, the same sgRNAs were removed from the testing datasets for CRISOT-FP. The overlapped sgRNAs are summarized in Supplementary Data 3. The CRISOT-FP models are trained on the Group I datasets, which are independent to the Group II datasets. Source data are provided as a Source Data file.

Supplementary Data 2). The sum of all SHAP importance values at each position indicated that the PAM-proximal positions were more important than the PAM-distal positions (Fig. 4a, left panel), which is consistent with previous studies on the molecular mechanism of Cas9[42,43]. The results also indicated that positions 5, 8 and 9 were more important among the PAM-distal positions, which was consistent with the previous study using a free energy model[44].

The interaction features were sorted by the sums at all positions (Fig. 4b). Among the top 5 features, E_surf (nonpolar solvation free energy), resA@N1 resB@N1 (the distance between N atoms on the bases of RNA and DNA nucleotides) and hbond_bp5 (hydrogen bonds) were related to the binding stability of the hybrid, indicating the importance of the binding stability to the activation of Cas9. These results were consistent with those in previous studies[14,43,45]. The curvature of the RNA-DNA hybrid is important to Cas9 and might affect the binding affinity of Cas9 with the RNA-DNA hybrid because the curvature of DNA was proven to influence the protein-DNA affinity[46]. We also summarized the contributions of different types of interaction features (Fig. 4c). The results showed that the hydrogen bonding feature contributed up to 4.3% (at position 12) to the model, even though the hydrogen bonding feature contained only one feature. The atom position and binding free energy features contributed averages of 54.1% and 23.5%, respectively, indicating the importance of the atom position features and the potential molecular mechanisms other than binding stability. In fact, we noticed that the positions of some atoms on the DNA nucleotide indicated by the distance and angle features (resB@C1' resB@N9, ang:resB@C1' resB@O5' resB@C3'_cos, ang:resB@O5' resB@N9 resB@C1'_cos, resB@P resB@C5' and resB@C5' resB@O5') were among the top 10 key features (Fig. 4b). The positions of these atoms might influence the interactions between the sgRNA-DNA hybrid and specific residues in Cas9, thus affecting the cleavage.

Finally, we used the key interaction features to design a flexible off-target scoring function, i.e., CRISOT-Score. For each position, we collected the top features (Supplementary Data 2) and their SHAP score values (Supplementary Data 2). These values were then aggregated to score specific bp types at specific positions. For example, to calculate the score of AA (sgRNA vs. off-target sequence) at position 5 using the top 5 key features, the mean SHAP score values related to the top 5 features at position 5 were collected, and their products were then aggregated as a final score (see Methods). We then determined how many features were needed to achieve considerable off-target prediction performance. Score maps were computed using the top 1 to the top 30 features at each position. The off-target scoring performances on different datasets were compared to those of the CRISOT-FP models (Fig. 4d and Fig. S13). As the feature number increased, the performance of CRISOT-Score increased sharply and achieved average ROC-AUC and PR-AUC scores of 99.2% and 76.0%, respectively, when the top 16 features were used. The ROC-AUC score remained at a high level, and the PR-AUC score slightly increased as the features increased. Therefore, we selected the top 24 features (Fig. S12) of each position to construct CRISOT-Score. The score map (Fig. S14) indicated that different positions favored different complementary bps. For example, positions 11 and 13 preferred AA over TT, positions 19 and 20 preferred TT over AA, and position 20 preferred CC over GG[47,48]. Different positions varied greatly in their sensitivity to mismatches. The PAM-distal positions, especially positions 1 and 2, were not sensitive to mismatches, whereas positions 13 to 17, which are within the seed region, were sensitive to mismatches.

**Comprehensive evaluations of CRISOT-Score.** The effectiveness of CRISOT-Score are evaluated comprehensively. In the first evaluation, CRISOT-Score was compared with 7 state-of-the-art hypothesis-driven off-target scoring methods including KinPred[34], CRISPRoff[5], uCRISPR[6],

MIT[7], CFD[8], CCTop[49] and CROPit[50] (learning-based methods were not included). We compared their performance on the Group II benchmark datasets (Table 1) with the measurements of ROC and PR (Fig. 4e, Fig. S15 and Fig. S16). CRISOT-Score significantly surpassed all seven existing off-target scoring methods in predicting off-target activities of all benchmark datasets ($p < 0.05$, Fig. S16). Of note, KinPred and CRISPRoff are mechanism-based scoring method that were derived from kinetic models[34] and the binding energy of the Cas9-sgRNA-DNA complex[5]. In comparison, CRISOT-Score included not only the binding free energy features, but also the atom position, hydrogen bonding and base pair/step geometric features that systematically reflect the interactions of the Cas9-sgRNA-DNA complex. Therefore, incorporating more details of the molecular interactions of Cas9 could benefit the prediction of off-target effects.

In the second evaluation, we tested how CRISOT-Score scores the active and inactive off-targets (Fig. 4f and Fig. S17). Among all potential off-target sites of the datasets, the experimentally validated off-target sites were defined as active off-targets, whereas the rests were inactive off-targets. The violin plots showed the CRISOT-Score distributions of the active vs. inactive off-targets (Fig. 4f). For the in vitro Change-seq, Site-seq and Circle-seq datasets, the CRISOT-Score of the active off-targets distributed generally distinct from the inactive off-targets. The peaks of the active off-targets were near the peaks of the inactive off-targets. In comparison, the CRISOT-Score of the active off-targets distributed greatly distinct from those of the inactive off-targets on the in cell Guide-seq and TTISS datasets. The peaks of the active off-targets of these datasets were approximately 0.7, and few active off-targets were scored less than 0.6. The results indicated that CRISOT-Score identified the active and inactive off-targets well, especially on the Guide-seq and TTISS datasets. In comparison, all of the other off-target scoring methods, but CRISPRoff, were not able to identify the active and inactive off-targets (Fig. S17).

In the third evaluation, we tested the proportions of active off-targets at different ranges of CRISOT-Score (Fig. 4g and Fig. S18). Generally, the frequency of active off-targets increased as the CRISOT-Score increased for all of the Group I and Group II datasets. The average frequencies summarized in Fig. 4g showed that the frequencies of active off-targets at CRISOT-Score ranges greater than 0.8 were higher than 0.9, and the frequencies of active off-targets at CRISOT-Score ranges lower than 0.5 were almost 0. It was indicated that a CRISOT-Score value that is higher than 0.8 almost indicates an active off-target, and a CRISOT-Score that is less than 0.5 almost indicates an inactive off-target. In comparison, CRISPRoff scores (Fig. S19) did not show a tendency that higher score sites were more likely to be active off-targets, and the frequency of active off-targets at the highest score range was 0, indicating its poorer scoring ability over CRISOT-Score.

## CRISOT-Spec provides effective sgRNA targeting specificity evaluation of Cas9 system

While CRISOT-Score evaluates the off-target effect of a specific off-target site, a specificity score is commonly designed to evaluate the specificity of a given sgRNA in terms of its genome-wide off-target profile. There are several CRISPR specificity scoring tools, e.g., MOFF-aggregrate[35], CRISPRspec[5], MIT[7] and CFD[8], which aggregate the off-target scores of the genome-wide off-target sites and calculate summary scores as the specificity scores. Thus, the existing specificity scores may relate to the number of potential off-target sites, i.e., a greater number of potential off-target sites usually results in a poorer specificity score. However, only a small fraction of the potential off-target sites are active off-target sites. As shown in Fig. 5a, the fractions of active off-targets were almost zero when the CRISOT-Score values were less than 0.5, while were almost one when CRISOT-Score higher than 0.85. Take the higher and lower specificity sgRNAs in Change-seq datasets (Fig. 5b and Fig. S20) as an example, although their total

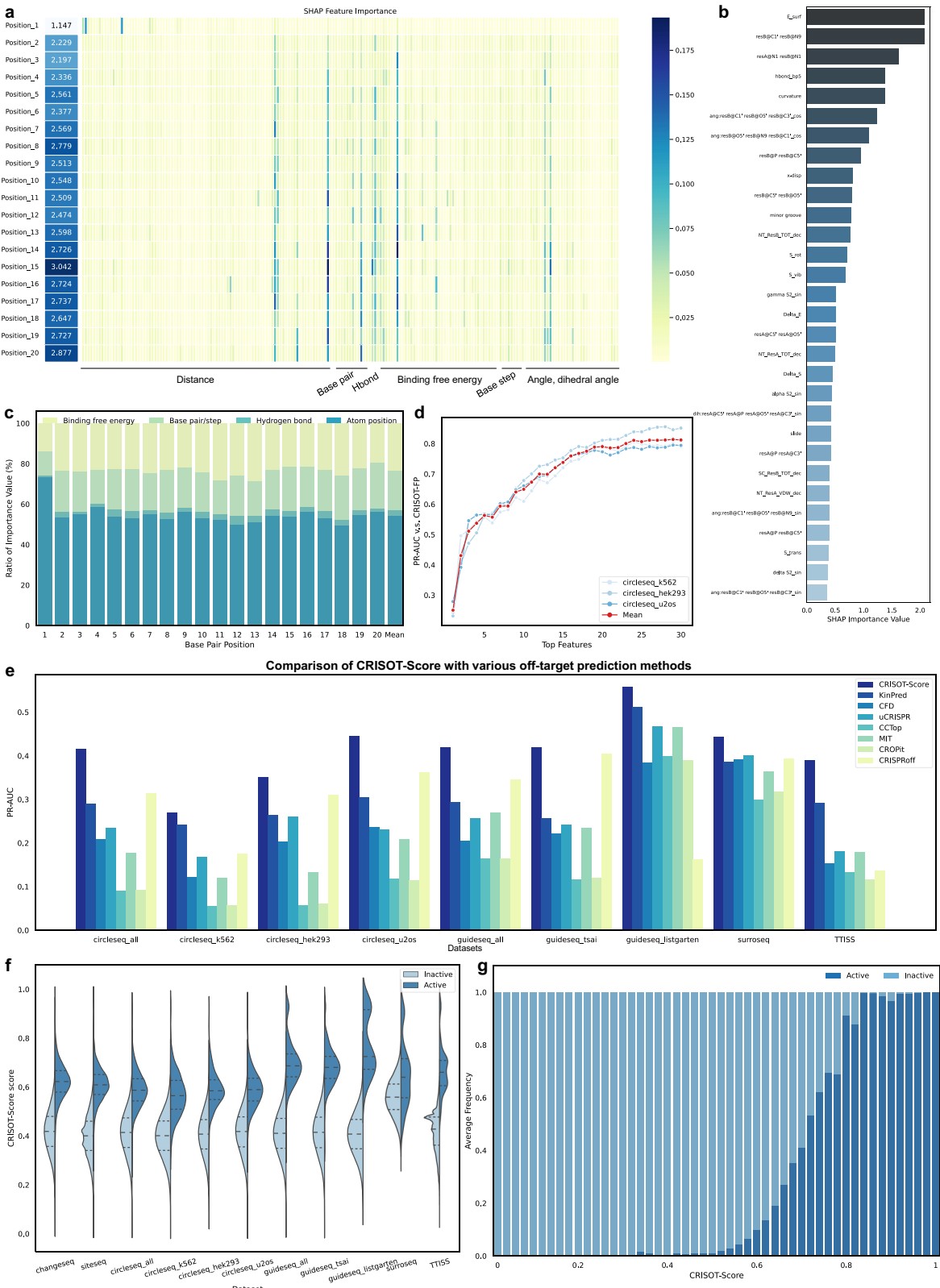

numbers of potential off-target sites were similar, the lower specificity sgRNAs have a greater number of off-target sites that were scored higher than 0.5. That meant severe off-target effects could occur if a significant portion of the potential off-target sites scored higher than 0.5, indicating that during the evaluation of the genome-wide off-target profile of a given sgRNA, more attention should be given to those with a higher CRISOT-Score.

To this end, we developed a specificity scoring method, i.e., CRISOT-Spec. For a given sgRNA, CRISOT-Spec searched its genome-wide off-target sites and calculated their CRISOT-Score values. The numbers of potential off-target sites at different ranges of CRISOT-Score values (Fig. 5b) were then calculated, and multiplied by the relative probability of being active off-targets (Fig. 5a). The product values were aggregated to derive the CRISOT-Spec values as a

**Fig. 4 | Feature importance analysis and the evaluation of CRISOT-Score.**
**a** Overall SHAP feature importance of the RNA-DNA molecular interaction features. The position importance values on the left are the sums of the rows on the right. **b** Top 30 important features. The importance values are the sums of the lines in (**a**). **c** The ratio of feature importance of various types of features on different positions. **d** PR-AUC Performances of CRISOT-Score developed using different numbers of key RNA-DNA molecular interaction features for each position. Results on the different benchmark datasets are compared to the results achieved by the CRISOT-FP models. **e** Comparison of CRISOT-Score with the state-of-the-art off-target scoring methods. The PR-AUC results indicate that CRISOT-Score surpasses the existing

off-target scoring methods. **f** Violin plots of CRISOT-Score scores on active and inactive off-target sites of different datasets. The numbers of active and inactive off-target sites can be found in Table 1. Among all potential off-target sites of the datasets, the experimentally validated ones were defined as active off-targets, and the rests were inactive off-targets. The three dashed lines in each violin show the quartiles of the data. **g** Frequencies of active and inactive off-target sites that are in different CRISOT-Score ranges. The presented result is averaged from Groups I and II benchmark datasets (Fig. S18, $n = 11$). Source data are provided as a Source Data file.

specificity score (see Methods). Because the probability of active off-targets with CRISOT-Score values lower than 0.5 was almost zero, CRISOT-Spec focused on off-target sites with CRISOT-Score values higher than 0.5, which were the most likely active off-targets.

To evaluate the performance of CRISOT-Spec, the CRISOT-Spec specificity score was compared to MOFF-aggregate, CRISPRspec, MIT and CFD specificity scores. The specificity scores were compared to the number of experimentally validated off-target sites in Groups I and II datasets. Off-target cleavage of a specific site would lead to great issues in clinical therapies, even when the cleavage efficiency is low. Thus, the number of validated off-target sites is important to evaluate the specificity of an sgRNA. CRISOT-Spec correlated well with the experimentally validated off-target sites (Fig. 5c) and achieved the best performance in predicting the off-target read fraction of the Site-seq dataset, with Spearman's correlation of −0.853. When compared with the other specificity score approaches, CRISOT-Spec significantly surpassed the CRISPRspec, MIT and CFD specificity scores ($p < 0.05$, Fig. 5d, e and Fig. S21), and was comparable to the only learning-based method, the MOFF-aggregate.

## CRISOT-Opti optimizes a sgRNA to reduce off-target effects while maintaining its on-target effects

Despite considerable on-target efficiency, an sgRNA may still have severe off-target effects (Fig. 5). In fact, the CRISPR system can tolerate a few mismatches, and the influence of a single mismatch on the targeting efficiency may be small[8,47]. In contrast, a single mutation in the sgRNA may significantly change the off-target profile and thus the specificity of a sgRNA. As the saying goes, "Better a diamond with a flaw than a pebble without". CRISOT-Opti was designed to find out such "a diamond with a flaw" - an sgRNA not perfectly match the target while present better targeting specificity. For a given sgRNA, CRISOT-Opti introduced 3 kinds of mutations to each of the 20 nucleotides (Fig. 5f). These mutated sgRNAs were paired with the target sequence to calculate the CRISOT-Score values. Although different types of mutations on the sgRNA generally decreased the CRISOT-Score value (Fig. S22), the results of CRISOT-Score indicated that a sequence with CRISOT-Score value of higher than 0.8 was always cleaved (i.e., active off-target) in most off-target detection experiments (Fig. S18), and a sequence with CRISOT-Score value of higher than 0.85 was cleaved in all of the off-target detection experiments. Therefore, to ensure considerable cleavage of the target DNA, we can set the threshold of CRISOT-Score to 0.8, 0.85 or higher, so that CRISOT-Opti only considered mutated sgRNAs with CRISOT-Score values higher than the threshold. Since the sgRNAs were mutated, the off-target profiles were changed. CRISOT-Opti calculated the CRISOT-Spec scores of the mutated sgRNAs based on their new off-target profiles. The scores were compared to the score of the wild-type (WT) sgRNA. Because higher CRISOT-Spec values indicate higher targeting specificity of an sgRNA, those mutated sgRNAs with increased CRISOT-Spec could be considered better substitutions for the WT sgRNA.

Take the GAGTCCGAGCAGAAGAAGAAGGG sequence on the human *EXM1* gene as an example of sgRNA optimization, which was

also investigated by previous Guide-seq study[37,38] (Fig. 5g, h). The WT sgRNA could cleave several off-target sites (CRISOT-Spec = 0.364), especially the three sites with CRISOT-Score values higher than 0.75. In this case, CRISOT-Opti mutated each nucleotide of the 20-nt sgRNA. Among a total of 60 mutated sgRNAs, 38 sgRNAs with CRISOT-Score values higher than 0.85 were chosen for the specificity evaluation (Supplementary Data 4). The CRISOT-Spec scores indicated that the A11 > C mutation (A at position 11 of the sgRNA was mutated to C) was the best substitution, of which the CRISOT-Spec score increased to 0.720, while the CRISOT-Score is still high at 0.867. In comparison, if the sgRNA was improperly mutated (e.g., C6 > A in Fig. S23), the targeting specificity would greatly decrease. Similarly, when targeting another site (e.g., GGGAAA-GACCCAGCATCCGTGGG in Supplementary Data 4), we also found an optimized substitution using CRISOT-Opti.

To validate the optimization result, we performed Guide-seq screening to detect genome-wide off-targets of the WT and A11 > C sgRNAs (Supplementary Data 5). Although the on-target efficiency was reduced when the mutated sgRNA was used instead of the WT sgRNA, the mutated sgRNA still maintained sufficient on-target efficiency (Fig. 5h, lower panel), which is consistent with the previous study that A11 > C sgRNA would maintain about 38% targeting activities[38]. On the other hand, the results showed that the A11 > C sgRNA greatly reduced off-target editing (Fig. 5h, i). The number of off-target reads was reduced from 51967 to 1072 by approximately 50 folds. The off-target read fraction was significantly reduced from 0.653 to 0.162, and the active off-target sites from 57 to 11. Most of the off-target sites were either eliminated or greatly reduced. For example, the off-target site on gene *HCN1*, which detected 35129 off-target reads when the WT sgRNA was used, detected a 281-fold reduction when the mutated sgRNA was used; and the off-target site on gene *SEMA5A*, which detected 3064 off-target reads when the WT sgRNA was used, was totally eliminated when the mutated sgRNA was used. The *HCN1* gene is involved in spontaneous rhythmic activity in both heart and brain[51], and the *SEMA5A* gene contributes to axonal guidance during neural development[52]. The optimized sgRNA well protected these genes to be targeted, which will greatly reduce side effects when using the CRISPR-Cas9 system in clinical trials.

## Experimental validation of sgRNA optimization for two important therapeutic genes *PCSK9* and *BCL11A* using CRISOT

In this study, we further performed additional proof-of-concept experimental validation to demonstrate the effectiveness of applying CRISOT for off-target evaluation and sgRNA optimization in gene therapy. To this end, two important therapeutic genes, *PCSK9* and *BCL11A*, are tested here. CRISPR has been used to knockdown the PCSK9 gene in primates to lower the level of cholesterol[53], and the editing of the *BCL11A* gene to treat transfusion-dependent β-thalassemia and sickle cell disease are in clinical trials[54]. The safety of CRISPR editing on these genes are of high concern. As a proof-of-concept example, we used CRISOT to evaluate the off-target profile of the gene editing and optimize two sgRNAs that have unsatisfying specificities targeting the *PCSK9* and *BCL11A* genes. We performed whole genome screening (WGS) experiment, an unbiased and direct method for

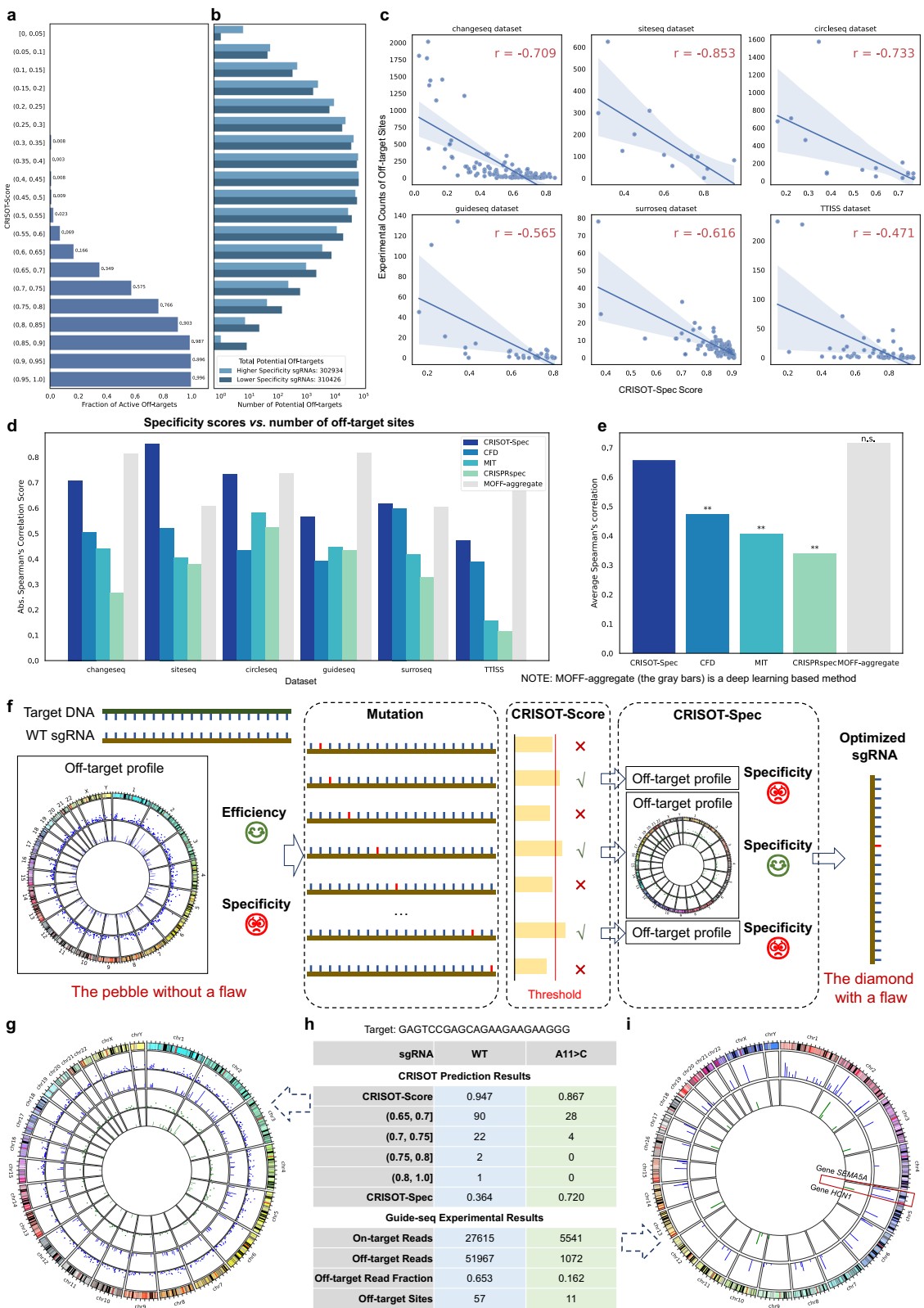

assessing off-target effects of Cas9, on the HEK293T cell line to experimentally validate the off-target profiles of sgRNAs.

We selected two optimized sgRNAs for *PCSK9* (G10 > T mutation) and *BCL11A* (G11 > T) using CRISOT, with CRISOT-Score scores of about 0.8 (Fig. 6a and Fig. S24). We performed WGS experiments and found that the off-target effects of the optimized sgRNAs were greatly reduced, while the on-target frequencies were maintained in a

considerable level (Fig. 6a, Fig. S25 and Supplementary Data 6). Notably, the on-target frequency (0.351) of the optimized sgRNA for *PCSK9* was comparable to that of the WT sgRNA (on-target frequency = 0.365). Collectively, this study indicates that CRISOT can be used for optimized sgRNA design to improve the gene editing safety, while maintaining the on-target efficacy of CRISPR therapeutics.

**Fig. 5 | CRISOT-Spec for scoring the targeting specificity and CRISOT-Opti for sgRNA optimization. a** Probabilities of being active off-target sites. The probabilities of different CRISOT-Score ranges are shown as the average fractions of active off-targets in the benchmark datasets. **b** Numbers of potential off-target sites in different CRISOT-Score ranges. We sort the 110 sgRNAs of Change-seq dataset according to the numbers of potential off-target sites and divide them into 11 groups to minimize the difference of numbers of potential off-target sites in each group. Top and bottom three sgRNAs with the least and most experimentally validated off-target sites are selected as the higher and lower specificity sgRNAs, respectively, for each group. The presented result is a summary of the 2-nd to the 10-th groups. **c** Spearman's correlations between the CRISOT-Spec scores and the experimentally validated numbers of off-target sites reported by the different experiments. The error bands represent the confidence intervals of 95% for the regression estimates. **d, e** Comparison of CRISOT-Spec with existing sgRNA specificity evaluation tools for predicting the number of off-target sites. Absolute Spearman's correlation scores are shown. The gray bars are results of MOFF-

aggregate, a learning-based method. **e** The results of one-sided paired t-test (n.s.: Not significant, $p > 0.05$; **$p < 0.01$; $n = 5$). The p-values are 0.007, 0.002, 0.001 and 0.757, respectively. **f** Conceptual framework of CRISOT-Opti optimization. **g** Comparison of off-target profiles of the WT and optimized sgRNAs. The circos plots show the off-target sites with CRISOT-Score values > 0.6. The scatter points indicate off-targets with CRISOT-Score values > 0.6, and the lines indicate those CRISOT-Score values > 0.65. The blue and green colors indicate the off-target sites of WT and optimized sgRNAs, respectively. The optimized sgRNA is optimized by setting a CRISOT-Score threshold to 0.85. **h** Summary of the CRISOT prediction results (higher panel) and the Guide-seq experimental results (lower panel). **i**, Comparison of experimental off-target profiles of the WT and optimized sgRNAs. The blue and green lines of the circos plot indicate the off-target sites of WT and optimized sgRNAs, respectively, which are detected by Guide-seq. Line length indicates the log10-transformed off-target reads. Source data are provided as a Source Data file.

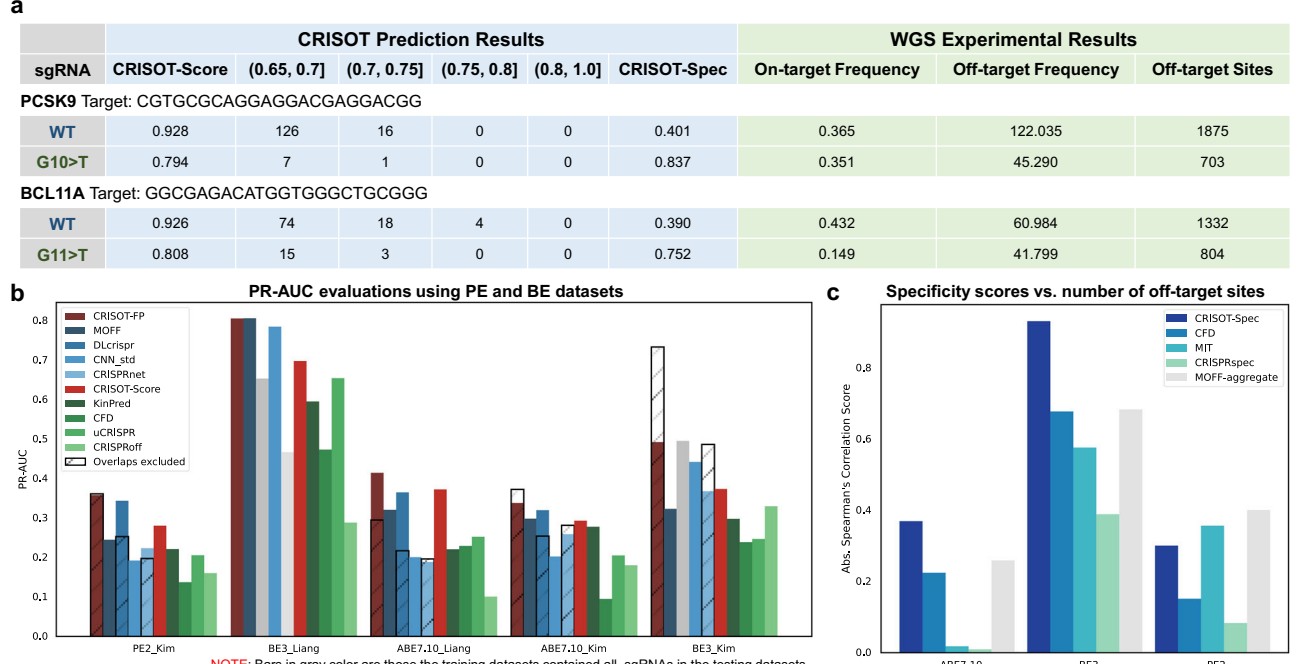

**Fig. 6 | Applications of CRISOT for therapeutic gene editing optimization and off-target prediction in BE and Prime editors. a** The CRISOT prediction results and WGS experimental results of wild-type and optimized sgRNAs targeting the PCSK9 and BCL11A genes. **b** Comparison of CRISOT-FP and CRISOT-Score with the existing off-target prediction methods for predicting the off-target effects of base and prime editors using PR-AUC. Learning-based methods are colored in blue, hypothesis-driven methods are colored in green. Bars in gray color are those the training datasets contained all sgRNAs in the testing datasets. Bars with slash hatch are presented by excluding the overlapped sgRNAs. Of note,

for fair comparisons, the same sgRNAs were removed from the testing datasets for CRISOT-FP. The overlapped sgRNAs are summarized in Supplementary Data 3. The CRISOT-FP models are trained on the Group I datasets, which are independent to the Group III datasets. **c** Comparison of CRISOT-Spec with existing sgRNA specificity evaluation tools for predicting the number of off-target sites detected in base and prime editing. The gray bars are results of MOFF-aggregate, a learning-based method. Source data are provided as a Source Data file.

## CRISOT facilitates genome-wide off-target prediction for base and prime editors

Base (BEs) and prime editors (PEs) enabled precise editing of the target sequence without DNA double-strand breaks (DSBs), while they also involve the interaction of RNA-DNA hybrid. Although not quite proper, off-target prediction methods of Cas9 were usually used to evaluate the off-target effects of BEs and PEs including BE3, ABE7.10 and PE2 etc. Here, we curated Group III (PE2[55], ABE7.10[56,57] and BE3[57,58]) off-target datasets (Table 1, see Methods), and evaluate the performance of CRISOT in predicting the off-target effects and targeting specificities of BEs and PEs[59] since CRISOT captures the common mechanisms of

these various CRISPR systems by characterizing the interaction of RNA-DNA hybrid. The Group III datasets were independent to the training (Group I) datasets.

Using these datasets, we found that CRISOT achieved better performance than the compared methods in predicting the off-target effects of BEs and PEs as expected (Fig. 6b and Fig. S26, S27). For the learning-based models, although they achieved comparable performance with CRISOT-FP on the ROC-AUC results, CRISOT-FP outperformed MOFF, CNN_std and CRISPRnet models on the PR-AUC results. The CRISOT-FP also surpassed DLcrispr after removing the overlapped sgRNAs to avoid data leakage of DLcrispr ($p < 0.05$,

Fig. 6b and Fig. S27). For the hypothesis-driven models, CRISOT-Score surpassed the existing hypothesis-driven off-target scoring methods in terms of PR-AUC and ROC-AUC ($p < 0.05$, Fig. 6b and Fig. S27). We also tested the performance of CRISOT-Spec in predicting the targeting specificities of BEs and PEs (Fig. 6c and Fig. S28). CRISOT-Spec achieved the best performance in predicting the targeting specificity of the ABE7.10 and BE3 datasets. Although the performance of MIT and MOFF-aggregate specificity scores were higher than CRISOT-Spec in the PE dataset, the MOFF-aggregate is a learning-based method and CRISOT-Spec greatly surpassed them in the ABE7.10 and BE3 datasets. Collectively, the application of CRISOT in BEs and PEs indicated that CRISOT achieved good generalizability in the prediction of the off-target effects and targeting specificities of BEs and PEs.

## Discussions

The molecular mechanism of Cas9-mediated DNA cleavage is driven by the interactions of the Cas9-sgRNA-DNA complex[14,15]. In this study, we systematically analyzed the RNA-DNA molecular interaction features using MD simulations, and the CRISOT suite was developed based on the RNA-DNA molecular interaction fingerprints. The CRISOT suite is designed as a generalizable system for off-target prediction, targeting specificity evaluation and sgRNA optimization by the integration of artificial intelligence (AI) and MD technology. In summary, CRISOT can be taken as a one-stop platform to greatly improve in silico sgRNA design and reduce off-target effect for various CRISPR system by investigating the molecular interaction mechanism.

Various applications were presented to demonstrate the utility of CRISOT. The CRISOT suite was used to optimize two sgRNAs targeting the *PCSK9* and *BCL11A* genes. The WGS off-target detection experiments validated the optimization results and indicated that CRISOT could help users to find an optimized sgRNA with almost no sacrifice of on-target efficacy. In addition, the CRISOT suite can also be used to evaluate the off-target effects of BEs and PEs, indicating the generalizability of CRISOT among distinct CRISPR systems.

Despite the success of CRISOT, several improvements are expected in future work. First, this study only considered off-targets with NGG PAMs and without DNA/RNA bulges. Apart from the canonical NGG PAM, Cas9 has been reported to cleave off-target sequences with alternative PAM sequences, such as NAG and NGA[60], although the cleavage rates were much smaller than those of the canonical PAM sequences. Future work should take into account various PAM sequences and DNA/RNA bulges. Second, our RNA-DNA molecular interaction features and CRISOT-FP could provide a deep understanding of the molecular mechanism of Cas9; however, the molecular mechanism of Cas9, especially the allosteric and cleavage mechanisms, contains molecular interactions that could not be reflected by the RNA-DNA hybrid. More comprehensive simulations and investigations on the Cas9-sgRNA-DNA complex are still required. Thirdly, features reflecting the target accessibility, e.g., chromatin-opening information and DNA methylation information, would improve the performance if they were properly augmented, however, such information are not guaranteed to be obtained. Therefore, we did not consider such information in this version to make CRISOT tools generally usable. Fourthly, we trained XGB models because they are simple and powerful, suitable for training tabular data, and provide great interpretations of the model. Various complex machine learning algorithms, especially deep learning algorithms can be incorporated in CRISOT in a flexible way in the future. Finally, the validations of our study are mainly performed in silico in the current stage and the experimental validation is served as a complementary. Additional large-scale experimental validations on sgRNAs targeting important therapeutic genes using Guide-seq, Change-seq, WGS etc. will make the results more convincing, while it is waiting to be performed in the future.

## Methods

### Preparation of RNA-DNA hybrids

We designed a of RNA-DNA hybrids containing 256 RNA-DNA hybrids (Fig. 2a) to calculate the RNA-DNA interaction features. Sixteen types of RNA-DNA base pairs, including matched and mismatched ones, were designed for feature calculation. Two additional base pairs were added up-and down-stream of the base pair (4 * 4 types, matched only), forming 4*16*4 = 256 types of 3-mer RNA-DNA hybrids for feature calculation to make the resulted features more representative. Finally, because the terminal base pairs were more likely to break during the simulations, we added 3-mer padding base pairs (rG-dC, the most stable base pair) up-and down-stream of the core base pairs to stabilize the simulation system. The original structure of the 9-bp RNA-DNA duplex was extracted from a high-resolution crystal structure of the Cas9-sgRNA-DNA ternary complex (PDB [5y36])[13]. We used the web 3DNA platform[61] to perform the three steps of nucleic acid base mutations to construct initial RNA-DNA hybrid structures for MD simulations. First, the original sequence was mutated to construct a (r5'-GGG-GGG-GGG)(d3'-CCC-CCC-CCC) hybrid (GGG-CCC, which indicates the 3 bps in the middle). Then the fourth and sixth bps were mutated, and various Watson–Crick bps were maintained, to construct 16 DGD-RCR hybrids (D and R represent DNA and RNA nucleotides, respectively). Finally, the fifth bp was mutated, and various nucleic acid bases were generated to construct a total of 256 9-bp hybrids. Short MD simulations (10 ns) were performed to optimize the structures before and after the mutations. The constructed hybrid structures varied only in the 4th-6th bps, in which only the middle bps contained bp mismatches.

### Molecular dynamics simulations

The all-atom RNA-DNA hybrids were solvated in TIP3P explicit water, and Na+ and Cl- ions were added to neutralize the systems. The DNA strands adopted the AMBER/parmbsc1 force field[62], whereas the RNA strands adopted Shaw's force field[63]. The combination of the parmbsc1 force field and Shaw's force field demonstrated good quality in MD simulations of RNA-DNA hybrids[64]. All MD simulations were performed with the GROMACS 2016 package[65]. The systems were first prepared by energy minimization, followed by 1 ns NVT and 1 ns NPT simulations with the RNA-DNA hybrids fixed. The systems were then equilibrated by 20 ns NPT simulations. The production simulations were performed at 310 K for 100 ns. The simulation time step was set to 2 fs, and the simulation trajectories were collected every 10 ps. Electrostatic interactions were calculated using the PME algorithm, with a cutoff radius of 1.2 nm. The temperature coupling was calculated using a Berendsen thermostat, with a coupling time of 0.1 ps. Lincs constraints were applied to all bonds. The production simulations were performed for 3 structures randomly chosen from the equilibration simulations. Therefore, the final features for each type of base pair were calculated based on 16*3*100 = 4800 ns of MD simulations. The MD simulations were performed using Xeon E5-4640 CPU on the National Supercomputer Center in Guangzhou, China. The computational time for a single production run was about 20.5 hours, and the total computational time for all simulation systems was about 15,744 h.

### MD trajectories analysis

Three replications of the MD trajectories of each RNA-DNA hybrids were analyzed to calculate the interaction features. An analysis of the base pair (e.g., shear, stretch, and stagger) and base step (e.g., twist, rise, and slide) parameters was performed using the do_x3dna package and dnaMD Python module[66]. The atom position features represented by distances, angles and dihedral angles of atoms were calculated using pytraj, a python package of CPPTRAJ[67] from AmberTools. P, C5', C3', O5', C1, N9 and N1 atoms and P, C5', C3', O5', C1, N1 and N3 atoms were selected for purines and pyrimidines, respectively (Fig. S29). The position parameters were calculated using the pytraj packages.

Hydrogen bonds were calculated using the GROMACS package[65]. The binding free energies and free energy decompositions of the RNA-DNA hybrids were calculated using Molecular mechanics/Generalized-Born Surface Area (MM-GBSA) method using the gmx_MMPBSA[40] program. The gmx_MMPBSA program is a GROMACS tool based on AMBER's binding free energy calculation engine to perform free energy calculations with GROMACS files.

## Feature engineering

The feature values varied between different types of bps, but were similar when the nearest bps changed (Fig. S1). However, some feature values may vary greatly among the different nearest bps (e.g., CA's hydrogen bonds in Fig. S2a). Therefore, we compared the correlation among features derived from different nearest bps (Fig. S1, right panels). The boxplots of Pearson's correlation scores of all features indicated that the mean values correlated best to different types of nearest bps (Fig. 2b). The mean values were calculated to represent the 16 bp types. These features were then engineered. For angles, dihedral angles and other angle features in the base pair/step geometric features, the values might not properly represent their differences. To this end, we took the sin and cos values of the angles instead, and the other features were scaled to a range of −1 to 1. Variances of the features were calculated and the features with variances less than 0.01 were eliminated. The feature engineering resulted in a total of 193 features (Supplementary Data 1), including 1 hydrogen bonding feature, 42 binding free energy features, 114 atom position features and 36 base pair/step geometric features, which were used to encode sgRNA-DNA hybrids (Fig. 2c).

## CRISPR off-target benchmark datasets

We constructed nine off-target benchmark datasets that contain data detected by different experiments (Table 1). The first two datasets (Group I) are in vitro datasets, namely the Change-seq and Site-seq datasets, which contain sgRNAs that are different from each other, are developed for training. The Change-seq dataset[68] contains 110 sgRNAs accounting for 21545 validated off-targets. The Site-seq dataset[14] contains 12 sgRNAs accounting for 2047 validated off-targets.

The second group of datasets (Group II) contained four datasets that are independent to the Group I datasets, namely the Circle-seq, Guide-seq, Surro-seq and TTISS datasets, are used as independent testing datasets to validate the models. The in vitro Circle-seq dataset[60], is comprised of Circle-seq data of three cell types, i.e., the K562 dataset containing 6 sgRNAs and 751 off-targets, HEK293 dataset containing 4 sgRNAs and 857 off-targets, and U2OS dataset containing 5 sgRNAs and 2844 off-targets. The in cell Guide-seq dataset, is comprised of Guide-seq data from two independent studies, i.e., Tsai's dataset containing 10 sgRNAs and 341 off-targets[37] and Listgarten's dataset containing 23 sgRNAs and 53 off-targets[38]. The Surro-seq dataset[39] is a targeted in cell dataset that contains 105 sgRNAs and 714 validated off-targets. Because the cutoff of fold-change (FC) would change the sensitivity of the Surro-seq dataset, we set 6 FC cutoffs (2, 4, 8, 16, 32, and 64), and ignored those sites with $p$-values > 0.05, which resulted in various datasets with different sensitivity (validated/potential ratios: 819/6540, 549/6741, 371/6763, 263/6764, 207/6764 and 174/6764, respectively). The in cell TTISS dataset[40] contains 59 sgRNA and 866 off-targets.

The last three datasets (Group III) contained in vitro off-target datasets of base and prime editors, which are independent from the Group I datasets, were used as testing datasets for further validations. The PE2 dataset[55] contains 9 sgRNAs accounting for 1493 validated off-targets. The BE3 dataset is comprised of Kim subset[58] containing 7 sgRNAs accounting for 76 validated off-targets and Liang subset containing 2 sgRNAs accounting for 3 off-targets. The ABE7.10 dataset is comprised of Kim subset[56] containing 7 sgRNAs accounting for 212

validated off-targets and Liang subset[57] containing 6 sgRNAs accounting for 44 off-targets.

The genome-wide off-target loci with a maximum of six nucleotide mismatches were calculated. The benchmark datasets represent different experimental genome-wide CRISPR off-target detection techniques. The experimentally validated off-target sites were defined according to the methods described in the corresponding articles. For example, Change-seq is a highly sensitive in vitro technique, thus, only those data with more than 100 Change-seq reads were considered as active off-targets, following the research of Change-seq[68]. The Cas-Offinder program[69], which is a versatile tool that searches all potential off-target sites, was used to search for genome-wide off-target sequences with a maximum of 6 mismatches (with no bulges). Because CRISOT-FP did not encode the PAM sequence, we considered only off-target sites with NGG PAMs. Models are trained on Group I datasets, and tested using Groups II and III datasets. All datasets are available in Zenodo repository[70]. The compared learning-based methods were trained on various datasets that were different from CRISOT-FP. The overlapped sgRNAs between the datasets in this study and the training datasets used by the other learning-based methods were summarized in Supplementary Data 3.

## Building machine learning models for CRISPR off-target prediction based on CRISOT-FP

The XGB algorithm is based on tree boosting[33], which is a fast and scalable learning algorithm and has shown great ability in feature interpretation. We used the XGB classifier to train leave-ones-out models, and predicted off-target effects of independent datasets using the average scores of the trained models. The XGB models used gbtree method as booster, and used logistic regression for binary classification. For benchmark purposes, we also trained supporting vector machine (SVM), logistic regression (LR) and random forest (RF) algorithms models using the scikit-learn package. The SVM classifier models used radial basis function (rbf) kernel, the LR classifier models used limited-memory BFGS (lbfgs) solver, and the RF classifier models used the Gini impurity as criteria. The detailed hyperparameter tuning and configurations are shown in Supplementary Data 7.

## Feature importance analysis in CRISOT-FP

The feature importance was calculated by the tree SHAP algorithm[41], which interprets feature importance scores from the tree-based XGB models. SHAP is a unified approach that can be used to explain the output of the XGB models[71]. SHAP feature importance values were calculated as the mean absolute values of the SHAP scores for each XGB model. The presented SHAP feature importance values were the combination of the SHAP importance values of the models trained on the three datasets. Analysis were performed on models trained on the Change-seq and Site-seq datasets. The SHAP importance values are shown as features vs. positions (Supplementary Data 2). The SHAP importance values of different positions were the sums of all the importance values at different positions. SHAP importance values of different features were the sums of importance values at all 20 positions.

## Design of CRISOT-Score

We selected the top $N$ features of each position to calculate the CRISOT-Score values. For a bp $RD$ ($R$ denotes the representation of sgRNA sequence (U - > T) and $D$ denotes the DNA sequence) at position $i$, its score was computed as

$$S_{RD,i} = \sum_{j=1}^{N} \mathrm{mean}\left(\left[V_i^{F_{RD,j}}{}_1, V_i^{F_{RD,j}}{}_2, \ldots, V_i^{F_{RD,j}}{}_n\right]\right) \quad (1)$$

where $F_{RD,j}$ is the top j-th feature; $V_i^{F_{RD,j}}{}_n$ is the n-th SHAP score values for $F_{RD,j}$ at position i. The $\mathrm{mean}([V_i^{F_{RD,j}}{}_1, V_i^{F_{RD,j}}{}_2, \ldots, V_i^{F_{RD,j}}{}_n])$ values are

summarized in Supplementary Data 2. The score of a pair of target and off-target sequences can be calculated by aggregating the scores at the 20 positions.

$$S = \sum_{i=1}^{20} S_{RD,i} \qquad (2)$$

Finally, we introduce two constants $a$ and $b$ to make that the final score is in the range of [0, 1]. Therefore, the CRISOT-Score value is computed as

$$CRISOT - Score = aS + b \qquad (3)$$

An off-target sequence with a higher CRISOT-Score value means that it is more likely to be cleaved.

### Design of CRISOT-Spec
CRISOT-Spec collects all of the off-target sites that are the most likely to be cleaved and aggregates their probabilities of being active off-targets. An off-target site is rarely cleaved if its CRISOT-Score value is less than 0.6 (Fig. 5a). For a given sgRNA, CRISOT-Spec calculates the CRISOT-Scores values of all of the potential off-target sites (which were determined by Cas-Offinder with no more than 6 mismatches). CRISOT-Spec then counts the numbers of off-target sites that are scored in the ranges presented in Fig. 5a, and multiplies them by their probabilities of being active off-targets. The aggregated off-target probability is the sum of the products.

$$P_{off} = \sum_{k=1}^{20} N_k P_k \qquad (4)$$

where $N_k$ and $P_k$ are the number of potential off-target sites and the off-target probability of the $k$-th range. Finally, the CRISOT-Spec value is calculated as

$$CRISOT - Spec = \frac{10}{10 + P_{off}} \qquad (5)$$

This equation was modified from the MIT specificity score[7]. The range of the CRISOT-Spec score is (0, 1]. An sgRNA with a higher CRISOT-Spec score has higher specificity across the whole genome. For comparison, we rebuilt the CRISPRspec[5], MIT[7] and CFD[8] specificity scores. The MIT and CFD specificity scores were rebuilt based on the CRISPOR package[72].

### Design of CRISOT-Opti
The CRISOT-Opti is based on the CRISOT-Score and CRISOT-Spec values. For a given sgRNA, CRISOT-Opti mutates each of the nucleotides at the 20 positions of the sgRNA and calculates the CRISOT-Score values of the mutated sgRNAs vs. the target DNA. The mutated sgRNAs with CRISOT-Score values higher than 0.8 are considered to be satisfactory with considerable cleavage and are chosen for further evaluation, whereas those with CRISOT-Score values lower than 0.8 were not used for further evaluation. Then, CRISOT-Spec values were computed to sort the mutated sgRNAs. The high-ranking sgRNA is the optimized sgRNA with the least off-target effects. A higher CRISOT-Score threshold allows for a more reliable sgRNA substitution with more reliable on-target efficiency. Furthermore, several additional rules, *e.g.*, the targeting activity of a mutated sgRNA, can be set to achieve a more reliable substitution.

### Guide-seq off-target detection
We used Guide-seq, a powerful off-target detection technique, to detect the genome-wide off-target sites of both WT and optimized sgRNAs[37]. HEK293 cells (SCSP-5209, obtained from Cell bank of

Shanghai Institute of Biochemistry and Cell biology, Chinese Academy of Sciences) were used for Guide-seq experiments. HEK293 cells were cultured in Dulbecco's modified Eagle medium (DMEM, Gibco) supplemented with 10% FBS (BI) and 1% penicillin/streptomycin (Beyotime) at 37 °C in 5% $CO_2$ incubators. Two complementary oligonucleotides (oGS1 and oGS2, Table S1) was used to generate the standard Guide-seq double-stranded oligodeoxynucleotide (dsODN). Cas9 and sgRNA encoding plasmids, and dsODN were transfected into the HEK293 cells using electroporation method. Genomic DNA was then isolated and the completed Guide-seq library was quantified by qPCR. After sequencing, the sequencing data were analyzed based on the Guide-seq analysis package (https://github.com/tsailabSJ/guideseq). The off-target sites (Supplementary Data 5) with no more than 6 mismatches were mapped to human genome reference (hg38).

### Whole genome sequencing (WGS) off-target detection
U6-gRNA scaffold-chicken $_\beta$-actin-SpCas9-CMV-EGFP (Donated by Hui Yang lab) was used as the backbone. The annealed sgRNAs (Fig. 6a) were inserted into the backbone plasmid using T4 ligase kit (Vazyme). HEK293T cells (SCSP-502, obtained from Cell bank of Shanghai Institute of Biochemistry and Cell biology, Chinese Academy of Sciences) were cultured in DMEM supplemented with 10% FBS and 1% penicillin/streptomycin at 37 °C in 5% $CO_2$ incubators. The SpCas9-sgRNA plasmids were transfected using polyethyleneimine (PEI, Polyscience) according to the manufacturer's protocols. 48 h after transfection, cells were washed with PBS and digested with 0.25% trypsin (YEASEN). Then cells were filtered with a 40 μm cell strainer. The EGFP positive cells of 1 million were sorted by flow cytometer (BD FACS AriaI III). Genomic DNA was extracted using TIANamp Genomic DNA Kit (TIANGEN) according to the manufacturer's protocols. Nest primers were designed around target sites, and the inside PCR products were sent to Azenta for sanger sequencing. The sequences of nest primers used in the WGS experiments were listed in Table S2. Then 2 μg of each sample was prepared for whole-genome sequencing in Azenta.

After sequencing, the sequencing data were analyzed. The off-target sites with no more than 6 mismatches were mapped to human genome reference (hg38). The analysis results were the combination of three independent replications for each sgRNA. Off-target sites with total reads less than 10, or with background INDEL frequency greater than 50% were excluded. Off-target frequencies were fixed using the background INDEL frequencies, and those with fixed frequencies greater than 0.01 were considered as off-target sites (Supplementary Data 6).

### Statistics & reproducibility
Sample sizes were determined based on literature precedence for genome editing experiments. No data were excluded from the analyses. The experiments were not randomized. The Investigators were not blinded to allocation during experiments and outcome assessment.

### Reporting summary
Further information on research design is available in the Nature Portfolio Reporting Summary linked to this article.

## Data availability
Source data are provided with this paper. All data supporting the findings of this study are available in the paper and the Supplementary Information files. Additional original data that support the findings are available from the corresponding author upon request. The Guide-seq sequencing data for sgRNAs targeting *EMX1* generated in this study have been deposited in the NCBI Sequence Read Archive under accession number PRJNA785744, and the WGS data for sgRNAs targeting *PCSK9* and *BCL11A* generated in this study have been deposited under accession number PRJNA921906. Supplementary data for the

features, feature importance values, overlapped sgRNAs, sgRNA optimization, off-target data, et al. are available as Supplementary Data 1–7. The training/testing datasets have been deposited in the Zenodo repository[70]. Source data are provided with this paper.

## Code availability

All code of the CRISOT suite is available at https://github.com/bm2-lab/CRISOT or via Zenodo[73]. The web server of CRISOT is available at https://crisot.aigene.org.cn/.

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

## Acknowledgements

Q.L., Q.Z., Q.C.C., and G.H.C. were supported by Roche pRED Informatics Advanced Analytics Postdoctoral Fellowship Program (aligned with the Roche pRED Postdoctoral Fellowship Program RPF-500). Q.L. was supported by the National Key Research and Development Program of China (Grant No. 2021YFF1201200, No. 2021YFF1200900), National Natural Science Foundation of China (Grant No. 31970638 and 61572361), Shanghai Natural Science Foundation Program (Grant No. 17ZR1449400), Shanghai Artificial Intelligence Technology Standard Project (Grant No. 19DZ2200900), Shanghai Shuguang scholars project, WeBank scholars project and Fundamental Research Funds for the Central Universities. E.W.Z., Q.C.C., and G.H.C. were supported by the National Natural Science Foundation of China (Grant No. 32371549, 62102286 and 62002265, respectively). E.W.Z. was supported by the Science Technology and Innovation Commission of Shenzhen Municipality of China (ZDSYS 20200811142605017). The computational calculations were performed in the National Supercomputer Center in Guangzhou and the Intelligence Computing Data Reactor in Zhejiang lab.

## Author contributions

Q.L., Q.Z. and E.W.Z. conceived, designed, and supervised this study. Q.C.C. performed molecular simulations and analysis. Q.C.C. and G.H.C. performed machine learning and analysis. Q.C.C., H.H.Z., G.H.C., W.N.L., W.H.L., and J.Y.W. performed further analysis and experimental validation. Q.C.C., J.T., L.W.D., and H.G. developed the package and website. Q.C.C., G.H.C. and H.H.Z. wrote the manuscript. All authors commented on the final manuscript.

## Competing interests

Q.C.C., G.H.C., Q.Z., and Q.L. have filed a patent application (PCT/EP2022/087932, public) on the CRISOT suite, the patent applicants are F. Hoffmann-La Roche AG (for all designated States except USA), Hoffmann-La Roche Inc. (for USA only) and Tongji University (all designated States). Q.Z. was formerly employed by Roche. The remaining authors declare no competing interests.
