## [Peer Review File · Nature Communications]

Reviewers' Comments:

Reviewer #1:

Remarks to the Author:

The manuscript entitled "Genome-wide CRISPR off-target prediction and optimization using molecular interaction fingerprints", by Q. Chen and co-authors, reports on a new computational methodology to predict genome-wide off-target effects of genome editing with CRISPR-Cas9. The work is based on a complex methodology that uses molecular simulations, machine learning and experimental validation. The results indicate convincing improvement with respect to existing methods.

Let me specify that I am not an expert in all parts of the work. I can only assess technically the simulation part and all physicochemical aspects.

With this limiting premise, I find the whole work very well designed and carried out. The study addresses a hot topic in current biochemical research and deserves wide attention. The prediction of off-target effects is the only possible route toward using CRISPR-based systems for medical therapies. As such, it is of crucial importance.

Let me start by commenting on the **simulation part**, which is my specialty.

There is nothing that I would change. The design of sequences, the mutations, the initial equilibration and production runs all look appropriate. The choice of force field is also accurately tailored. There are only three points that the authors could consider for minor revision:

1. Motivate the choice of 9-mers, as opposed to longer sequences. And why 256 sequences?
2. Specify in the text which computer was used for the simulations and the total computational time (CPU/GPU time)
3. 100 ns is nowadays considered short for simulations of DNA; on the other hand, I agree that it is a reasonable compromise for simulating so many structures. It would be nice if the authors could add a comment about the suitability of this time duration. The only convincing way to do so, is to run longer (1 microsecond) simulations for 2-5 randomly chosen structures and report negligible differences in the predicted features.

About **feature selection**, which must be based on the physicochemical properties of the systems and is also my expertise.

1. It is unclear to me how the free energy was estimated from the MD simulations, please specify.
2. I think it would also be good to write some more details about the specification of the atom position features. I guess they are for a selection of atoms, not all atoms. Are they coordinates? Or relative coordinates?
3. Also specify the software or protocol used to compute the base pair/step geometric features. Is it Curves? Or 3DNA? Or something else?
4. Figure S1 is barely intelligible. Since it is a supporting element, I recommend to break it down into 5 figures, so that the values can be read, as well as the number of rows and columns. The caption could be longer and more

descriptive.

5. On pg. 5 the authors mention 193 features. On pg. 18 they mention 246 features. What is the correct number? It's probably contained in Table S1, but I did not count the columns.... Actually it would be more effective to change the rows to columns and columns to rows in Table S1, so that the number of features is obvious from the number of rows.

At a more general level, my **greatest concern** about this manuscript is the imprecise use of the term "molecular interaction features". In particular, the expression "*To fully characterize the molecular interaction features of the CRISPR system*" is problematic. In fact, the CRISPR-Cas9 complex has different components and a full characterization of the molecular interaction features would require simultaneous treatment of DNA-Protein, DNA-DNA, DNA-RNA and RNA-protein interactions. Instead, the authors include in their training only features that characterize the DNA-RNA interaction.

Major revisions.

1. The authors should change the language throughout the manuscript to reflect the limitation on the molecular interactions included in their study. They should always use the term "RNA-DNA molecular interactions" instead of generic "molecular interactions".
2. The authors address the issue of off-target effects by a possible strategy that is the optimization of the sgRNA. This is not the only strategy that has been proposed. At least another strategy is by protein mutations to modify molecular interactions. Thus, the authors should include in the introduction a comment on the fact that their approach is a possibility, but not the only possibility, and give credit to other strategies by citing appropriate references, both experimental and computational.

Reviewer #2:

Remarks to the Author:

Chen et al., presents an exciting framework for CRISPR off-target and analysis. In the work the authors introduce molecular dynamics as part of the sgRNA – DNA interactions and do subsequently extract features which together with a boosting algorithm is used to predict off-target activities. The authors show further with experimental follow up analysis on two gRNAs how the introduction of mismatches in the on-target gRNA can lead to a strong reduction in the off-target activity. Furthermore, the authors show the applicability to base editors and prime editors. Overall the paper is well organized, but in a number of instances not completely clear, which potentially raises some concerns and hampers a full assessment.

In the LGO test set (l 179-181) precisely what is the criteria to define when “different off-target sequences” for the same sgRNA are actually different? And what is the justification for the criteria?

Although the authors use cas-offinder it is unclear if the authors allow and evaluate on bulged (and not same length interacting) sgRNA and target DNA? The authors should make this clear and if this is a limitation in their method then say so. For example CRISPRnet claim to be able to handle such data. Is that included in the method comparison?

In several instances the performances of the methods compared is small and the authors should therefore use a test e.g. that of Steiger to test for the significance in differences in performances between the methods.

In the comparison to other methods (Fig 2f-k) have the authors made sure that none of the methods compared have been trained on the data used for testing? For example comparing on guide-seq is that a subset that was not used for training some of the methods? This is obviously critical in making a fair comparison. The authors should clarify this and provide sets of gRNAs in supplementary material on precisely which off-target for which gRNAs the methods are compared on and flag the subsets used in the respective trainings.

Since the authors use molecular dynamics, I wonder why the authors are not comparing to kinetics base methods such as Eslami-Mossallam et al., NCOMM, 2022 and Fu et al., NCOMM, 2022. The latter also considers sequence determinants for CRISPR specificity which seems to be highly relevant to include in the authors study. The authors should take these methods into account in their work.

Furthermore, do the authors in anyway take into account that some data are in-vivo and others in-vitro and hence as indicated in the literature impact the off-targets e.g. by methylated DNA. The authors should elaborate.

For the identification of the PAM distal position I wonder if there is an overlap to what has already been observed in the on-target analysis by Corsi et al. NCOMM 2022. The authors should compare.

For the feature extraction, the authors seems to preclude them from extracting the sgRNA-DNA binding energy, although it was identified as the most important feature in the on-target analysis (Xiang et al., NCOMM 2021) also explicitly incorporated into the prediction model. It makes sense to compare the most important features to the binding energy and discuss similarities and differences. How important is the RNA:DNA binding energy and which features extracted maybe be underlying the binding energy?

For data shown in Fig 3g (1) precisely how are “active and inactive off-targets” defined? (2) I also wonder how this correspond to corresponding considerations of the energy-based analysis on on-targets presented by Corsi et al.? Can you authors elaborate? (3) Precisely how many data points are in the active/inactive groups for each dataset?

For the third evaluation (from I 338) it is still unclear how a CRISOT-Score less than 0.6 can indicate inactive off-targets (is that referring to the observations 367-369?)

What precisely do the authors mean by “A CRISOT-Score value of higher than 0.8 was unacceptable”. Please elaborate.

What does it mean to be cleaved. How precisely is that defined for the different data sets?

Concerning the CRISOT-spec measure. At a glance it makes sense, but since the authors use CRISOT-score to trained on all off-target data, there is potential for data leakage. To test the robustness, the authors should test the CRISOT-spec on off-targets which have not at all been used in the training.

For the CIRSOT-Opti, this seems to be somewhat similar to make changes to an on-target while ensuring its binding energy is within the sweet-spot binding interval. It would be interesting to hear the authors perspectives on this.

Minor

The blue scale coloring in the performance figures e.g. fig 2 makes it very hard to see the individual methods. Please use a more reader friendly color scale (check for example color brewer)

L330 Fig 6f → Fig 3g ??

legends missing in Fig S19.

Reviewer #3:

Remarks to the Author:

The authors prepared 193 molecular interaction features per base pair of sgRNA-DNA, generating $193 \times 20 = 3,860$ features, which were named CRISOT-FP. The use of CRISOT-FP led to higher machine learning performances compared to *Crista_feat*, one-hot, and two-hot encoding methods. Next, the authors generated models using XGB and CRISOT-FP and found that their models showed higher performance than the other pre-existing models including three other machine-learning-based models. Subsequent SHAP analysis revealed the importance of binding stability. The authors also developed CRISOT-Score, which was based on the top 15 features of each position. CRISOT-Score outperformed all previously reported off-target scoring methods. Furthermore, CRISOT-Score was useful in distinguishing the active vs. inactive off-targets. The authors subsequently developed CRISOT-Spec, which is the aggregation of CRISOT-Score values multiplied by the relative probability of being active off-targets. Additionally, the authors also developed CRISOT-Opti, which identifies sgRNAs that match the target sequences with only one-nucleotide mismatch and have much fewer off-target effects. The authors validated the usefulness of their models using example sgRNAs targeting PCSK9 and BCL11A. In addition, they demonstrated that their models are useful in predicting off-target effects for prime editing and base editing.

The study was well-designed and conducted. However, due to the limited novelty and the incremental improvement in the prediction performance, the manuscript would be appropriate for a more specialized journal such as communications biology.

Major comments

1. The authors utilized features related to molecular dynamics, including binding energy, for developing off-target prediction models. However, binding energy has been used previously (e.g. Alkan et al., "CRISPR-Cas9 off-targeting assessment with nucleic acid duplex energy parameters" (Genome Biology, 2018; reference 5 in this manuscript) and Pan et al., "Massively targeted evaluation of therapeutic CRISPR off-targets in cells" (Nature Communications, 2022; not cited in the current manuscript)). Thus, the novelty of the current manuscript is limited.
2. Regarding comment 1, how important are the other features besides binding energy? A quantitative assessment of the importance of these other features would strengthen the manuscript.
3. The improvement in off-target effect prediction performance is incremental. Considering molecular dynamics might not be a game-changer.
4. The authors showed the utility of CRISOT-Score in predicting the active and inactive off-targets (Fig. 3g, h). However, this evaluation was not performed in comparison with other state-of-the-art methods. The manuscript would be strengthened if the authors compare CRISOT-Score with the other methods in predicting active vs. inactive off-targets.

Minor comments

1. On page 9, "The violin plots showed the CRISOT-Score distributions of the active vs. inactive off-targets (Fig. 6f).". I believe this citation of Fig. 6f is an error.
2. Throughout the manuscript, the text in figures is often too small to read (e.g. Fig. 4e).
3. On the second paragraph of page 3, it is written as "genome-wide off-target sequencing including Guide-seq, Change-seq, Circle-seq etc.", but other than the three papers mentioned, no other databases were used. Therefore, the "etc." should be removed.
4. To help readers reproduce the results described in the manuscript, the authors should summarize all sgRNA sequences and target sequences from Guide-seq, Change-seq, and Circle-seq used in the machine learning process, along with the energy calculation values, key feature calculation values, all feature values utilized in the development of the CRISOT-FP machine learning model, and model prediction values, and present this information in a table.
5. In Fig. 2f, g, h, i, j, and k, 11 computer models were compared; however, in some figures, only 8 comparison bars are shown. Please ensure that all 11 models are shown in all relevant figures. Additionally, in Fig. 3e and 3f, Fig. 4d and 4e, and Fig. 5c, please include a comparison of all 11 computer models rather than comparing only some of them.
6. Column names in Table S1 are abbreviated, making it difficult to understand. Please provide a detailed explanation of the abbreviations used in the column names within the table legend.
7. Please include data from the circle-seq and change-seq datasets in addition to the GUIDE-seq dataset when presenting the data in Fig. 3h and Fig. 4, as the number of sgRNAs is too small (only 39 sgRNAs) (Table 1). The author could provide these results as Supplementary Information.
8. In Fig. 4b, only the results of two guide RNAs are shown, making it difficult to generalize the findings. It is recommended to present data for all guide RNAs.
9. On page 11, it is written, "CRISOT-Spec correlated well with the off-target read fraction for all Guide-seq sgRNAs (Fig. 4c)." However, the values at the bottom right of Fig. 4C (Guideseq-Listgarten dataset) are low, with -0.302 and -0.250. It is difficult to conclude that the model performs well based on this generalization. It is recommended to improve the model's performance and also note that there is a small amount of data for generalization. Moreover, it is suggested to include data from the Circle-seq and Change-seq datasets when presenting the results.
10. On page 15, Result 6 is described as an in vitro experiment. However, after reviewing the Methods section, it was found that the cells were transfected with sgRNA and Cas9, and genomic DNA was extracted for off-target analysis using whole genome sequencing, which might not be able to identify low-frequency off-target effects.

11. In Fig. 5a, only four sgRNAs were tested for validation purposes, which is an insufficient number. It is recommended to test at least 30 sgRNAs for validation. If whole genome sequencing is too expensive, it is suggested to use more commonly used methods such as Guide-seq, Change-seq, or Circle-seq, which are described in other papers, for validation experiments.

We would like to appreciate the reviewers for their careful reading of our paper, and for their thoughtful comments and insightful suggestions to improve our manuscript. We genuinely enjoyed incorporating the reviewers' comments, and the manuscript has been carefully revised (colored in **red**) in response to the comments.

Summary of the major changes:

(1) Datasets collection: We have collected additional off-target datasets and divided all datasets into 3 groups: the Group I datasets were independent from each other, which were used for training, validation and testing; the Group II datasets were used as independent testing datasets; the Group III datasets, which were off-target data of base and prime editors, were used for further evaluation on the generalizability of CRISOT. New models were trained on the Group I datasets.

(2) Methods comparison: Following the reviewers' comments, we have adopted MOFF (Eslami-Mossallam et al., NCOMM, 2022) and KinPred (Fu et al., NCOMM, 2022) as comparison methods. CRISOT showed better performance over the MOFF and KinPred models and the other state-of-the-art methods.

(3) Benchmark validation: Firstly, we have used additional independent datasets for validation. Secondly, following the Reviewer 2's comment, we have performed paired t-test to test for the significance in differences in performances between the methods. Finally, we have presented the results of all comparison methods, and indicated in the figures the potential for data leakage of some learning-based methods.

(4) Literal expression: Following Reviewer 1's comment, we have used the term “RNA-DNA molecular interactions” instead of generic “molecular interactions” to reflect the limitation on the molecular interactions included in their study. The title of this manuscript was also changed to "Genome-wide CRISPR off-target prediction and optimization using RNA-DNA interaction fingerprints".

(5) Methods description: Following the reviewers' comments, we have added some detailed descriptions on the methods, especially for the simulations and feature selection.

Our point-to-point responses can be found below. For clarity, the reviewers' comments are colored in **black** and our responses are in **blue**.

Reviewer #1 (Remarks to the Author):

The manuscript entitled “Genome-wide CRISPR off-target prediction and optimization using molecular interaction fingerprints”, by Q. Chen and co-authors, reports on a new computational methodology to predict genome-wide off-target effects of genome editing with CRISPR-Cas9. The work is based on a complex methodology that uses molecular simulations, machine learning and experimental validation. The results indicate convincing improvement with respect to existing methods.

Let me specify that I am not an expert in all parts of the work. I can only assess technically the simulation part and all physicochemical aspects.

With this limiting premise, I find the whole work very well designed and carried out. The study addresses a hot topic in current biochemical research and deserves wide attention. The prediction of off-target effects is the only possible route toward using CRISPR-based systems for medical therapies. As such, it is of crucial importance.

Let me start by commenting on the simulation part, which is my specialty.

There is nothing that I would change. The design of sequences, the mutations, the initial equilibration and production runs all look appropriate. The choice of force field is also accurately tailored. There are only three points that the authors could consider for minor revision:

1. Motivate the choice of 9-mers, as opposed to longer sequences. And why 256 sequences?

Authors' reply: Thanks for this question. We designed the simulation system to calculate the features of the RNA-DNA base pair (16 types including matched and mismatched ones). Two additional base pairs were added up-and down-stream of the base pair ($4 * 4$ types, matched only), forming $4 * 16 * 4 = 256$ types of 3-mer RNA-DNA hybrids for feature calculation to make the resulted features more representative. Finally, because the terminal base pairs were more likely to break during the simulations, we added 3-mer padding base pairs (rG-dC, the most stable base pair) up-and down-stream of the core base pairs to stabilize the simulation system.

We have added the following in the Methods section for clarity: "We designed a of RNA-DNA hybrids containing 256 RNA-DNA hybrids (Fig. 2a) to calculate the RNA-DNA interaction features. Sixteen types of RNA-DNA base pairs, including matched and mismatched ones, were designed for feature calculation. Two additional base pairs were added up-and down-stream of the base pair ($4 * 4$ types, matched only), forming $4*16*4=256$ types of 3-mer RNA-DNA hybrids for feature calculation to make the resulted features more representative. Finally, because the terminal base pairs were more likely to break during the simulations, we added 3-mer padding base pairs (rG-dC, the most stable base pair) up-and down-stream of the core base pairs to stabilize the simulation system."

2. Specify in the text which computer was used for the simulations and the total computational time (CPU/GPU time)

Authors' reply: Thanks! We have added the following in the Methods section to specify the computer and the total computational time of the MD simulations: " The MD simulations were performed using Xeon E5-4640 CPU on the National Supercomputer Center in Guangzhou, China. The computational time for a single production run was about 20.5 hours, and the total computational time for all simulation systems was about 15744 hours."

3. 100 ns is nowadays considered short for simulations of DNA; on the other hand, I agree that it is a reasonable compromise for simulating so many structures. It would be nice if the authors could add a comment about the suitability of this time duration. The only convincing way to do so, is to run longer (1 microsecond) simulations for 2-5 randomly chosen structures and report negligible differences in the predicted features.

Authors' reply: We thank the reviewer for this comment. We agree that 100-ns MD simulation is nowadays considered short for nucleotides, and we have considered two aspects to make the features derived from the MD simulations more convincing. On the one hand, we have run 100 ns production simulations for 3 structures randomly chosen from the 20-ns equilibration simulations. On the other hand, we designed $4*4=16$ hybrids for each of the 16 types of matched and mismatched base pairs. Therefore, the final features were calculated based on $16*3*100=4800$ ns of MD simulations. We have added the following to the Methods section to comment the suitability of time duration: "The production simulations were performed for 3 structures randomly chosen from the equilibration simulations. Therefore, the final features for each type of base pair were calculated based on $16*3*100=4800$ ns of MD simulations."

About feature selection, which must be based on the physicochemical properties of the systems and is also my expertise.

1. It is unclear to me how the free energy was estimated from the MD simulations, please specify.

Authors' reply: Thanks! We have added the following description on the calculation of binding free energies in the Methods section: "The binding free energies and free energy decompositions of the RNA-DNA hybrids were calculated using Molecular mechanics/Generalized-Born Surface Area (MM-GBSA) method using the gmx_MMPBSA program. The gmx_MMPBSA program is a GROMACS tool based on AMBER's binding free energy calculation engine to perform free energy calculations with GROMACS files."

2. I think it would also be good to write some more details about the specification of the atom position features. I guess they are for a selection of atoms, not all atoms. Are they coordinates? Or relative coordinates?

Authors' reply: Thanks for this comment. The atom position feature are relative coordinates represented by distances, angles and dihedral angles. The selection of atoms and the calculation of these features were given in the Methods section: "The atom position features represented by distances, angles and dihedral angles of atoms were calculated using pytraj package. P, C5', C3', O5', C1, N9 and N1 atoms and P, C5', C3', O5', C1, N1 and N3 atoms were selected for purines and pyrimidines, respectively (Fig. S21). The position parameters were calculated using the pytraj packages."

3. Also specify the software or protocol used to compute the base pair/step geometric features. Is it Curves? Or 3DNA? Or something else?

Authors' reply: Thanks! We used the 3DNA software to compute the base pair/step geometric features. The descriptions were presented in the Methods section: "An analysis of the base pair (e.g., shear, stretch, and stagger) and base step (e.g., twist, rise, and slide) parameters was performed using the do_x3dna package and dnaMD Python module."

4. Figure S1 is barely intelligible. Since it is a supporting element, I recommend to break it down into 5 figures, so that the values can be read, as well as the number of rows and columns. The caption could be longer and more descriptive.

Authors' reply: We thank the reviewer for the comment. We have broken down the Figure S1 into 5 figures.

5. On pg. 5 the authors mention 193 features. On pg. 18 they mention 246 features. What is the correct number? It's probably contained in Table S1, but I did not count the columns.... Actually it would be more effective to change the rows to columns and columns to rows in Table S1, so that the number of features is obvious from the number of rows.

Authors' reply: Thanks! Sorry for the confusion. We have calculated 246 draft features and finally adopted 193 features after feature engineering and filtering. We now deleted the descriptions on the 246 draft features for clarity. The Table S1 have been changed as the reviewer commented.

At a more general level, my greatest concern about this manuscript is the imprecise use of the term "molecular interaction features". In particular, the expression "To fully characterize the molecular interaction features of the CRISPR system" is problematic. In fact, the CRISPR-Cas9 complex has different components and a full characterization of the molecular interaction features would require simultaneous treatment of DNA-Protein, DNA-DNA, DNA-RNA and RNA-protein interactions. Instead, the authors include in their training only features that characterize the DNA-RNA interaction.

Major revisions.

1. The authors should change the language throughout the manuscript to reflect the limitation on the molecular interactions included in their study. They should always use the term “RNA-DNA molecular interactions” instead of generic “molecular interactions”.

Authors’ reply: We thank the reviewer for the comment. We have used the term “RNA-DNA molecular interactions” instead of generic “molecular interactions” in this article.

2. The authors address the issue of off-target effects by a possible strategy that is the optimization of the sgRNA. This is not the only strategy that has been proposed. At least another strategy is by protein mutations to modify molecular interactions. Thus, the authors should include in the introduction a comment on the fact that their approach is a possibility, but not the only possibility, and give credit to other strategies by citing appropriate references, both experimental and computational.

Authors’ reply: We agree with the reviewer's comment. Additional off-target reducing strategies includes protein engineering, delivering additional sgRNAs and so on. We have added the following in the Introduction section: " In addition, the previous off-target reducing strategies, e.g., protein engineering¹ and delivering additional sgRNAs², are complex and require additional experimental works, indicating a requirement for a simple yet powerful strategy to improve targeting specificity. ".

Reviewer #2 (Remarks to the Author):

Chen et al., presents an exciting framework for CRISPR off-target and analysis. In the work the authors introduce molecular dynamics as part of the sgRNA – DNA interactions and do subsequently extract features which together with a boosting algorithm is used to predict off-target activities. The authors show further with experimental follow up analysis on two gRNAs how the introduction of mismatches in the on-target gRNA can lead to a strong reduction in the off-target activity. Furthermore, the authors show the applicability to base editors and prime editors. Overall the paper is well organized, but in a number of instances not completely clear, which potentially raises some concerns and hampers a full assessment.

In the LGO test set (1 179-181) precisely what is the criteria to define when “different off-target sequences” for the same sgRNA are actually different? And what is the justification for the criteria?

Authors’ reply: Thanks! Sorry for the confusion. The leave-group-out (LGO) and leave-sgRNAs-out (LSO) splits can be illustrated by **Fig. R1** (corresponding to **Fig. S3**). A sgRNA (sgRNA_n) leads to j_n potential on/off-target sequences $[S_1^n, S_2^n, S_3^n, \dots, S_{j_n}^n]$. An input instance is a pair of sgRNA and one of the corresponding on-/off-target sequences (e.g. sgRNA_n and $S_{j_n}^n$). LGO test randomly held out 1/5 of the inputs as testing data, so that the training and testing contain all sgRNAs and different sequences of their corresponding on-/off-target sites. LSO test randomly held out 1/5 of the sgRNAs and the corresponding on-/off-target sequences as testing data, so that training and testing datasets contained different sgRNAs and off-target sequences. In this case, LSO is taken as a stricter and more challenging prediction task compared to LGO.

We have added **Fig. S3** to illustrate the LGO and LSO tests, and revised the text as following: “We performed two leave-ones-out tests to evaluate CRISOT-FP and the existing encoding methods (**Fig. S3**). The leave-group-out (LGO) test randomly held out 1/5 of the inputs as testing data, so that the training and testing datasets contain all sgRNAs and different sequences of their corresponding on-/off-target sites. The leave-group-out (LGO) test randomly held out 1/5 of the sgRNAs and the corresponding on-/off-target sequences as testing data, so that training and testing datasets contained different sgRNAs and off-target sequences. In this case, LSO is taken as a stricter and more challenging prediction task compared to LGO.”.

sgRNAs	On-/Off-Target Sequences																		
sgRNA ₁	S ₁ ¹	S ₂ ¹	S ₃ ¹	S ₁₋₂ ¹	S ₁₋₁ ¹	S ₁ ¹	LSO Training
sgRNA ₂	S ₁ ²	S ₂ ²	S ₃ ²	S ₁₋₂ ²	S ₁₋₁ ²	S ₁ ²		
sgRNA ₃	S ₁ ³	S ₂ ³	S ₃ ³	S ₁₋₂ ³	S ₁₋₁ ³	S ₁ ³		
sgRNA ₄	S ₁ ⁴	S ₂ ⁴	S ₃ ⁴	S ₁₋₂ ⁴	S ₁₋₁ ⁴	S ₁ ⁴		
...	...																		
sgRNA _{n-2}	S ₁ ⁿ⁻²	S ₂ ⁿ⁻²	S ₃ ⁿ⁻²	S ₁₋₂ ⁿ⁻²	S ₁₋₁ ⁿ⁻²	S ₁ ⁿ⁻²	LSO Testing	
sgRNA _{n-1}	S ₁ ⁿ⁻¹	S ₂ ⁿ⁻¹	S ₃ ⁿ⁻¹	S ₁₋₂ ⁿ⁻¹	S ₁₋₁ ⁿ⁻¹	S ₁ ⁿ⁻¹		
sgRNA _n	S ₁ ⁿ	S ₂ ⁿ	S ₃ ⁿ	S ₁₋₂ ⁿ	S ₁₋₁ ⁿ	S ₁ ⁿ		
LGO Training															LGO Testing				

sgRNA_n corresponds to j_n potential on/off-target sequences [S₁ⁿ, S₂ⁿ, S₃ⁿ, ... S_{j_n}ⁿ].
LGO: Leave-group-out
LSO: Leave-sgRNAs-out

Fig. R1 (Corresponding to **Fig. S3**) **Data splitting of the leave-group-out (LGO) and leave-sgRNAs-out (LSO) tests.** A sgRNA (sgRNA_n) leads to j_n potential on/off-target sequences [S₁ⁿ, S₂ⁿ, S₃ⁿ, ... S_{j_n}ⁿ]. An input instance is a pair of sgRNA and the on-/off-target sequence (e.g. sgRNA_n and S_{j_n}ⁿ). LGO test randomly held out 1/5 of the inputs as testing data, so that the training and testing contain all sgRNAs and different sequences of their corresponding on-/off-target sites. LSO test randomly held out 1/5 of the sgRNAs and the corresponding on-/off-target sequences as testing data, so that training and testing datasets contained different sgRNAs and off-target sequences.

Although the authors use cas-offinder it is unclear if the authors allow and evaluate on bulged (and not same length interacting) sgRNA and target DNA? The authors should make this clear and if this is a limitation in their method then say so. For example CRISPRnet claim to be able to handle such data. Is that included in the method comparison?

Authors' reply: Thanks! We did not consider DNA/RNA bulges in this study because no bulged structures were included in the MD simulations, and the derived RNA-DNA molecular interaction fingerprints were not able to represent DNA/RNA bulges. We agree with the reviewer that this is a limitation to be improved in future work. We have added a claim in the Methods section and added the following discussion in the Discussions section: "First, this study only considered off-targets with NGG PAMs and without DNA/RNA bulges. ... Future work should take into account various PAM sequences and DNA/RNA bulges. "

In several instances the performances of the methods compared is small and the authors should therefore use a test e.g. that of Steiger to test for the significance in differences in performances between the methods.

Authors' reply: We thank the reviewer for this comment. We have performed paired t-test to test for the significance in differences in performances between the methods, and added **Figures S9, S11** and **S27** in the supporting information.

In the comparison to other methods (Fig 2f-k) have the authors made sure that none of the methods compared have been trained on the data used for testing? For example comparing on guide-seq is that a subset that was not used for training some of the methods? This is obviously

critical in making a fair comparison. The authors should clarify this and provide sets of gRNAs in supplementary material on precisely which off-target for which gRNAs the methods are compared on and flag the subsets used in the respective trainings.

Authors' reply: We thank the reviewer for this comment. The Guide-seq and Circle-seq datasets were used in DLcrispr, CNN_std and CRISPRnet models, but our model still achieved better performance than these models in predicting most of these datasets. To make a fair comparison, we have added "#" in **Figures 2, 5, S9, S11 and S27** to indicate that the training data of these models contained sgRNAs in the testing datasets.

Since the authors use molecular dynamics, I wonder why the authors are not comparing to kinetics base methods such as Eslami-Mossallam et al., NCOMM, 2022 and Fu et al., NCOMM, 2022. The latter also considers sequence determinants for CRISPR specificity which seems to be highly relevant to include in the authors study. The authors should take these methods into account in their work.

Authors' reply: Following the reviewer's comment, we have adopted the MOFF (Eslami-Mossallam et al., NCOMM, 2022) and KinPred (Fu et al., NCOMM, 2022) off-target prediction methods as comparison methods in the validation studies.

Furthermore, do the authors in anyway take into account that some data are in-vivo and others in-vitro and hence as indicated in the literature impact the off-targets e.g. by methylated DNA. The authors should elaborate.

Authors' reply: Thanks for this valuable comment. The epigenetic features reflecting the target accessibility, e.g., DNA methylation, open chromatin, et al. were used in several previous studies, including in our former developed tool DeepCRISPR (Genome Biology 2018, 19:80). While the epigenetic impact on the off-targets remains unclear and it is waiting to be further investigated (e.g., Nature Biomedical Engineering 2018, 2:38–47). In addition, it is needed to specify such in vitro epigenetic information, which are not guaranteed to obtain. Therefore, in our study, to make the tool to be generally usable, we did not consider epigenetic information in this version.

We have added the following in the Discussions section: "Thirdly, features reflecting the target accessibility, e.g., chromatin-opening information and DNA methylation information, would improve the performance if they were properly augmented, however, such information are not guaranteed to obtain. Therefore, we did not consider such information in this version to make CRISOT tools generally usable".

For the identification of the PAM distal position I wonder if there is an overlap to what has already been observed in the on-target analysis by Corsi et al. NCOMM 2022. The authors should compare.

Authors' reply: Exactly! The results of important positions among the PAM-distal region in our study were consistent with the top positional nucleotide preferences found by Corsi et al. (NCOMM 2022).

We have added the following in Result 3.1 section: " The results also indicated that positions 5, 8 and 9 were more important among the PAM-distal positions, which was consistent with the previous study using a free energy model³".

For the feature extraction, the authors seems to preclude them from extracting the sgRNA-DNA binding energy, although it was identified as the most important feature in the on-target analysis (Xiang et al., NCOMM 2021) also explicitly incorporated into the prediction model. It makes sense to compare the most important features to the binding energy and discuss similarities and differences. How important is the RNA:DNA binding energy and which features extracted maybe be underlying the binding energy?

Authors' reply: We are sorry that we might have misled you for not including the binding energies in our features. In fact, the binding energies are included in the feature extraction (Fig. 1b) and they were among the top important features (Fig. 3b). The E_surf (nonpolar solvation free energy) was identified as the most important feature in this study, which is the nonpolar solvation free energy contribution of the binding free energy of RNA-DNA. However, the binding free energies were not the most contributive features (Fig. 3c). The atom position features contributed 54.1%, in comparison to 23.5% for the binding free energy features, indicating the importance of atom position features and the corresponding deeper molecular mechanism other than binding stability.

We have added the following descriptions in the Results 3.1 section: "E_surf (nonpolar solvation free energy), resA@N1 resB@N1 (the distance between N atoms on the bases of RNA and DNA nucleotides) and hbond_bp5 (hydrogen bonds) were related to the binding stability of the hybrid, indicating the importance of the binding stability to the activation of Cas9. ... The atom position and binding free energy features contributed averages of 54.1% and 23.5%, respectively, indicating the importance of the atom position features and the potential molecular mechanisms other than binding stability. "

For data shown in Fig 3g (1) precisely how are "active and inactive off-targets" defined? (2) I also wonder how this correspond to corresponding considerations of the energy-based analysis on on-targets presented by Corsi et al.? Can you authors elaborate? (3) Precisely how many data points are in the active/inactive groups for each dataset?

Authors' reply: We thank the reviewer for this comment.

(1) The "active and inactive off-targets" were defined according to whether they were experimentally validated. Among all potential off-target sites of the datasets, the experimentally validated ones were defined as active off-targets, and the rests were inactive off-targets.

(2) From the perspective of DNA cleavage, the on-targets presented by Corsi et al. were all "active" because they were "cleaved" even though the cleavage efficiencies of some on-targets were low. From the perspective of classification, the "active and inactive off-targets" were similar to the classification of "efficient and inefficient on-targets" by Corsi et al., but we used all of data points and did not consider the cleavage efficiencies. That means a potential off-target site is an "active off-target" if it is detected and validated by the off-target detection experiments, even though the cleavage efficiency may be considered "inefficient" in the study of Corsi et al.

(3) The numbers of data points in the active/inactive groups for each dataset can be found in Table 1.

In response to the reviewer's comment, we have made several revisions:

Section 3.2, "Among all potential off-target sites of the datasets, the experimentally validated off-target sites were defined as active off-targets, whereas the rests were inactive off-targets. ";

Fig. 3f, "Violin plots of CRISOT-Score scores on active and inactive off-target sites of different datasets. The numbers of active and inactive off-target sites can be found in **Table 1**. Among all potential off-target sites of the datasets, the experimentally validated ones were defined as active off-targets, and the rests were inactive off-targets."

For the third evaluation (from 1 338) it is still unclear how a CRISOT-Score less than 0.6 can indicate inactive off-targets (is that referring to the observations 367-369?)

Authors' reply: Sorry for the confusion. The results are relative to **Fig. R2** (corresponding to **Fig. 3g**). The previous version of this figure was summarized from the Guide-seq datasets, while the new version (**Fig. R2**) was summarized from Groups I and II benchmark datasets. According to Fig. R2, the frequency of active off-targets increased as the CRISOT-Score increased. The average frequencies of active off-targets at CRISOT-Score ranges greater than 0.8 were higher than 0.9, and the frequencies of active off-targets at CRISOT-Score ranges lower than 0.5 were almost 0. It was indicated that a CRISOT-Score value that is higher than

0.8 almost indicates an active off-target, and a CRISOT-Score that is less than 0.5 almost indicates an inactive off-target.

The former statement that "a CRISOT-Score less than 0.6 can indicate inactive off-targets" while "a CRISOT-Score value of higher than 0.8 was unacceptable" is now revised. We have revised the third evaluation in Section 3.2 as following: "Generally, the frequency of active off-targets increased as the CRISOT-Score increased for all of the Group I and Group II datasets. The average frequencies summarized in **Fig. 3g** showed that the frequencies of active off-targets at CRISOT-Score ranges greater than 0.8 were higher than 0.9, and the frequencies of active off-targets at CRISOT-Score ranges lower than 0.5 were almost 0. It was indicated that a CRISOT-Score value that is higher than 0.8 almost indicates an active off-target, and a CRISOT-Score that is less than 0.5 almost indicates an inactive off-target. "

Fig. R2 (corresponding to **Fig. 3g**) Frequencies of active and inactive off-target sites that are in different CRISOT-Score ranges. The presented result is averaged from Groups I and II benchmark datasets (**Fig. S18**, n=11).

What precisely do the authors mean by “A CRISOT-Score value of higher than 0.8 was unacceptable”. Please elaborate.

Authors’ reply: This comment is replied in the previous comment.

What does it mean to be cleaved. How precisely is that defined for the different data sets?

Authors’ reply: "Be cleaved" means that the off-target site was detected and validated by an off-target detection experiment, which was corresponding to the term "active off-target". The definition of validated off-target sites differs among different off-target detection methods. In this study, we defined them according to the methods described in the corresponding articles. We have added the following in the Methods section: " The experimentally validated off-target sites were defined according to the methods described in the corresponding articles. For example, Change-seq is a highly sensitive in vitro technique, thus, only those data with more

than 100 Change-seq reads were considered as active off-targets, following the description of Change-seq⁴. "

Concerning the CRISOT-spec measure. At a glance it makes sense, but since the authors use CRISOT-score to trained on all off-target data, there is potential for data leakage. To test the robustness, the authors should test the CRISOT-spec on off-targets which have not at all been used in the training.

Authors' reply: Thanks for this valuable comment. We have constructed additional independent datasets (the Group II datasets) for evaluation. The results showed that CRISOT-Spec significantly surpassed the CRISPRspec, MIT and CFD specificity scores (**Fig. 4d, e** and **Fig. S21**), and was comparable to the only learning-based method, the MOFF-aggregate.

For the CIRSOT-Opti, this seems to be somewhat similar to make changes to an on-target while ensuring its binding energy is within the sweet-spot binding interval. It would be interesting to hear the authors perspectives on this.

Authors' reply: Exactly. The CRISOT-Opti introduces mutation to the sgRNA while ensuring its CRISOT-Score higher than the threshold to ensure that the mutated sgRNA can successfully cleave the target DNA. This is similar to making changes to an on-target while ensuring its binding energy is within the sweet-spot binding interval. Furthermore, CRISOT-Opti aims to improve the targeting specificity, so an optimized sgRNA should not only hold a considerable CRISOT-Score score, but greatly increase its CRISOT-Spec score.

Minor

The blue scale coloring in the performance figures e.g. fig 2 makes it very hard to see the individual methods. Please use a more reader friendly color scale (check for example color brewer)

L330 Fig 6f → Fig 3g ??

legends missing in Fig S19.

Authors' reply: Thanks for this valuable comment. We have made revisions in the figures and tests.

Reviewer #3 (Remarks to the Author):

The authors prepared 193 molecular interaction features per base pair of sgRNA-DNA, generating $193 \times 20 = 3,860$ features, which were named CRISOT-FP. The use of CRISOT-FP led to higher machine learning performances compared to Crista_feat, one-hot, and two-hot encoding methods. Next, the authors generated models using XGB and CRISOT-FP and found that their models showed higher performance than the other pre-existing models including three other machine-learning-based models. Subsequent SHAP analysis revealed the importance of binding stability. The authors also developed CRISOT-Score, which was based on the top 15 features of each position. CRISOT-Score outperformed all previously reported off-target scoring methods. Furthermore, CRISOT-Score was useful in distinguishing the active vs. inactive off-targets. The authors subsequently developed CRISOT-Spec, which is the aggregation of CRISOT-Score values multiplied by the relative probability of being active off-targets. Additionally, the authors also developed CRISOT-Opti, which identifies sgRNAs that match the target sequences with only one-nucleotide mismatch and have much fewer off-target effects. The authors validated the usefulness of their models using example sgRNAs targeting PCSK9 and BCL11A. In addition, they demonstrated that their models are useful in predicting off-target effects for prime editing and base editing.

The study was well-designed and conducted. However, due to the limited novelty and the incremental improvement in the prediction performance, the manuscript would be appropriate for a more specialized journal such as communications biology.

Major comments

1. The authors utilized features related to molecular dynamics, including binding energy, for developing off-target prediction models. However, binding energy has been used previously (e.g. Alkan et al., "CRISPR-Cas9 off-targeting assessment with nucleic acid duplex energy parameters" (Genome Biology, 2018; reference 5 in this manuscript) and Pan et al., "Massively targeted evaluation of therapeutic CRISPR off-targets in cells" (Nature Communications, 2022; not cited in the current manuscript)). Thus, the novelty of the current manuscript is limited.

Authors' reply: Thanks for this comment. We agreed with the reviewer that the binding energy is an important feature and it is used in previous study, however, considering binding energy is not the only contribution in our study, while we aims at building a systematic framework for CRISPR off-target prediction and optimization by combining molecular simulations and AI techniques. Specifically, besides the consideration of binding energy, we would like to highlight the novelty and contributions of our study in the following aspects:

(1) In this study, we derived four types of RNA-DNA molecular interaction features from a series of molecular dynamics (MD) simulations, and developed the interaction fingerprints (CRISOT-FP) to train XGBoost classification models for off-target prediction. It can be seen that binding energy is one of the most important features for developing off-target prediction models, and has been used in many published off-target prediction models, which was also included in this study. However, our study indicated the other features, including hydrogen bond, atom position, base pair/step geometric features, also contributed greatly to the model. In fact, we calculated the contributions of the four types of features (**Fig. 3c**). The results showed that the atom position and binding free energy features contributed averages of 54.1% and 23.5%, respectively, indicating the importance of the atom position features and the potential molecular mechanisms other than binding stability.

(2) In addition to the novelty of MD-derived interactions and CRISOT-FP, we want to highlight the novelty of three related flexible modules based on CRISOT-FP, namely CRISOT-Score, CRISOT-Spec and CRISOT-Opti. a) CRISOT-Score was developed by identifying key features derived from CRISOT-FP models, which can quickly calculate the off-target score of a given pair of sgRNA and off-target sequences (**Fig. 1d**). b) CRISOT-Spec was developed to calculate the specificity score of a given sgRNA (**Fig. 1e**), by aggregating the CRISOT-Scores of the high-scored off-target sequences among all possible off-target sites. c) CRISOT-Opti was developed for sgRNA optimization (**Fig. 1f**). For an sgRNA with high editing efficiency and poor targeting specificity, CRISOT-Opti introduces a single nucleotide mutation for this given sgRNA by reducing the off-target effect of the given sgRNA while maintaining its on-target effect.

Taking together, a systematic and novel framework for CRISPR off-target prediction and optimization combining molecular simulations and AI techniques is designed in our study. The CRISPRoff off-target scoring method (Alkan et al., Genome Biology 2018) mentioned by the reviewer was one of the comparing methods adopted in our study, and the SURRO-seq dataset (Pan et al., Nature Communications 2022) was also adopted as a benchmark dataset in our study.

2. Regarding comment 1, how important are the other features besides binding energy? A quantitative assessment of the importance of these other features would strengthen the manuscript.

Authors' reply: Thanks for this valuable comment. We calculated the contributions of the four types of features (**Fig. 3c**). The results showed that the binding free energy features contributed 23.5% to the model, while the contribution of atom position features was up to 54.1%.

We have added the **Fig. 3c** and added the following results in the Results section: "We also summarized the contributions of different types of interaction features (**Fig. 3c**). The results showed that the hydrogen bonding feature contributed up to 4.3% (at position 12) to the model,

even though the hydrogen bonding feature contained only one feature. The atom position and binding free energy features contributed averages of 54.1% and 23.5%, respectively, indicating the importance of the atom position features and the potential molecular mechanisms other than binding stability.”.

3. The improvement in off-target effect prediction performance is incremental. Considering molecular dynamics might not be a game-changer.

Authors’ reply: Thanks! In response to the reviewer's comment to indicate the significant improvement achieved in our study, we have expanded the benchmark datasets (**Table 1**), performed additional benchmark evaluations (**Fig. 3f, g** and **Fig. 5b**), and, following Reviewer 2's comments, performed paired t-tests to compare the significance in differences in performances between CRISOT-FP and the other methods (**Figures S9, S11** and **S27**). As a result, the **Fig. R3** summarized the PR-AUC performances in Groups I-III datasets with paired t-tests (n.s.: not significant; *: $p < 0.05$; **: $p < 0.01$; ***: $p < 0.001$; ****: $p < 0.0001$). The results showed that only DLcrispr, whose training datasets contained sgRNAs in 9 out of the 11 testing datasets, achieved comparable performance in Group I and Group III datasets. CRISOT-FP significantly outperformed the other learning-based and hypothesis-driven methods, even though the training datasets of several methods contained sgRNAs in the testing datasets. These results indicated the significant improvement of CRISOT-FP in predicting the off-target effects, and the great generalization capability of CRISOT-FP.

Fig. R3 Summary of the PR-AUC results in Groups I-III datasets. a (corresponding to **Fig. S9c**) for Group I data sets, **b** (corresponding to **Fig. S11c**) for Group II datasets and **c** (corresponding to **Fig. S27b**) for Group III datasets. Paired t-test: n.s., not significant; *, $p < 0.05$; **, $p < 0.01$; ***, $p < 0.001$; ****, $p < 0.0001$.

4. The authors showed the utility of CRISOT-Score in predicting the active and inactive off-targets (Fig. 3g, h). However, this evaluation was not performed in comparison with other state-of-the-art methods. The manuscript would be strengthened if the authors compare CRISOT-Score with the other methods in predicting active vs. inactive off-targets.

Authors' reply: Thanks for this valuable comment. We have added the evaluations on the other comparison methods (**Fig. S17**). All of the other off-target scoring methods, except CRISPRoff, were not able to identify the active and inactive off-targets. Therefore, the scores of CRISPRoff were also evaluated in the third evaluation (**Fig. S19**). However, CRISPRoff scores (**Fig. S19**) did not show a tendency that higher score sites were more likely to be active off-targets, and the frequency of active off-targets at the highest score range was 0, indicating its poorer scoring ability compared to CRISOT-Score.

Minor comments

1. On page 9, "The violin plots showed the CRISOT-Score distributions of the active vs. inactive off-targets (Fig. 6f)." I believe this citation of Fig. 6f is an error.

Authors' reply: The typo has been corrected.

2. Throughout the manuscript, the text in figures is often too small to read (e.g. Fig. 4e).

Authors' reply: We thank the reviewer for this comment. We have checked the figures and reformatted the figures or increased the font size to make the texts readable.

3. On the second paragraph of page 3, it is written as "genome-wide off-target sequencing including Guide-seq, Change-seq, Circle-seq etc.", but other than the three papers mentioned, no other databases were used. Therefore, the "etc." should be removed.

Authors' reply: Thanks. Now we have expanded the benchmark datasets, the "etc." was kept.

4. To help readers reproduce the results described in the manuscript, the authors should summarize all sgRNA sequences and target sequences from Guide-seq, Change-seq, and Circle-seq used in the machine learning process, along with the energy calculation values, key feature calculation values, all feature values utilized in the development of the CRISOT-FP machine learning model, and model prediction values, and present this information in a table.

Authors' reply: Thanks for this comment. We have summarized the information of the benchmark datasets in Table 1, and all datasets were given along with the code of the package. The features, feature importance values, SHAP score values were given in Tables S1-S3.

5. In Fig. 2f, g, h, i, j, and k, 11 computer models were compared; however, in some figures, only 8 comparison bars are shown. Please ensure that all 11 models are shown in all relevant figures. Additionally, in Fig. 3e and 3f, Fig. 4d and 4e, and Fig. 5c, please include a comparison of all 11 computer models rather than comparing only some of them.

Authors' reply: Thanks for this comment. We did not show some results of the DLcrispr, CNN_std and CRISPRnet because the training datasets of these models contained sgRNAs that were in the testing datasets. Following the reviewer's comment, we have presented the results of all comparison methods and indicated the results in which the training datasets of these models contained sgRNAs that were in the testing datasets using "#" (**Fig. 2f, g, Fig. 5b, Fig. S9a, Fig. S11a and Fig. S27a**).

6. Column names in Table S1 are abbreviated, making it difficult to understand. Please provide a detailed explanation of the abbreviations used in the column names within the table legend.

Authors' reply: We have added detailed explanation of the feature names in **Table S1**.

7. Please include data from the circle-seq and change-seq datasets in addition to the GUIDE-seq dataset when presenting the data in Fig. 3h and Fig. 4, as the number of sgRNAs is too

small (only 39 sgRNAs) (Table 1). The author could provide these results as Supplementary Information.

Authors' reply: We have added the results on all of the Groups I and II datasets in **Fig. 3g**, **Fig. 4**, **Fig. S15** and **Fig. S21**.

8. In Fig. 4b, only the results of two guide RNAs are shown, making it difficult to generalize the findings. It is recommended to present data for all guide RNAs.

Authors' reply: Thanks for this comment. The result in **Fig. 4b** is convincing only when the numbers of potential off-target sites of two sgRNAs (or two groups of sgRNAs) were similar. Therefore, we used the Change-seq dataset, which contained 110 sgRNAs, to present this result. We sort the 110 sgRNAs of Change-seq dataset according to the numbers of potential off-target sites and divide them into 11 groups to minimize the difference of numbers of potential off-target sites in each group. Top and bottom three sgRNAs with the least and most experimental off-target sites are selected as the higher and lower specificity sgRNAs, respectively, for each group. The results (**Fig. S20**) showed that the numbers of potential off-target sites of the higher and lower specificity sgRNA varied greatly in the first and last groups, and were similar in groups 2-10. Therefore, we presented the results of the 2-nd to the 10-th groups in **Fig. 4b**.

9. On page 11, it is written, "CRISOT-Spec correlated well with the off-target read fraction for all Guide-seq sgRNAs (Fig. 4c)." However, the values at the bottom right of Fig. 4C (Guideseq-Listgarten dataset) are low, with -0.302 and -0.250. It is difficult to conclude that the model performs well based on this generalization. It is recommended to improve the model's performance and also note that there is a small amount of data for generalization. Moreover, it is suggested to include data from the Circle-seq and Change-seq datasets when presenting the results.

Authors' reply: Following the reviewer's comments, we have improved CRISOT-Spec and presented the results of all of the Groups I and II datasets (**Fig. 4c**).

10. On page 15, Result 6 is described as an in vitro experiment. However, after reviewing the Methods section, it was found that the cells were transfected with sgRNA and Cas9, and genomic DNA was extracted for off-target analysis using whole genome sequencing, which might not be able to identify low-frequency off-target effects.

Authors' reply: In Result 6, we intended to perform additional experiment (different from the Guide-seq used in the Result 5) to further validate the CRISOT-Opti. Although WGS has its limitations, it is an unbiased and direct method for assessing off-target effects of Cas9. Therefore, we believe that the results of WGS can be used for experimental validation in this study.

We have made the following revision in the Result 6 section: "In this study, we further performed additional experimental validation to demonstrate the effectiveness of applying CRISOT for off-target evaluation and sgRNA optimization in gene therapy. ... We performed whole genome screening (WGS) experiment, an unbiased and direct method for assessing off-target effects of Cas9, on the HEK293T."

11. In Fig. 5a, only four sgRNAs were tested for validation purposes, which is an insufficient number. It is recommended to test at least 30 sgRNAs for validation. If whole genome sequencing is too expensive, it is suggested to use more commonly used methods such as Guide-seq, Change-seq, or Circle-seq, which are described in other papers, for validation experiments.

Authors' reply: We agree with the reviewer that a greater number of experimentally validated sgRNAs will make the results more convincing. However, the validation of our study has been

mainly performed in-silico in the current stage, and the experimental validation is served as a complementary. Large-scale experimental validation are expected in the future.

We have added the following in the Discussions section: "Finally, the validation of our study are mainly performed in-silico in the current stage and the experimental validation is served as a complementary. Additional large-scale validation experiments on sgRNAs targeting important therapeutic genes using Guide-seq, Change-seq, WGS etc. will make the results more convincing, while it is waiting to be performed in the future."

References

1. Kleinstiver, B. P. *et al.* High-fidelity CRISPR-Cas9 nucleases with no detectable genome-wide off-target effects. *Nature* 529, 490–495 (2016).
2. Coelho, M. A. *et al.* CRISPR GUARD protects off-target sites from Cas9 nuclease activity using short guide RNAs. *Nat. Commun.* 11, 1–12 (2020).
3. Corsi, G. I. *et al.* CRISPR/Cas9 gRNA activity depends on free energy changes and on the target PAM context. *Nat. Commun.* 13, 1–14 (2022).
4. Lazzarotto, C. R. *et al.* CHANGE-seq reveals genetic and epigenetic effects on CRISPR-Cas9 genome-wide activity. *Nat. Biotechnol.* 38, 1317–1327 (2020).

Reviewers' Comments:

Reviewer #1:

Remarks to the Author:

The authors have exhaustively addressed all my comments and I recommend the current version of the manuscript for publication in *Nature Communications*.

I just wish to add an optional recommendation. It is related to my previous comment.

“The authors address the issue of off-target effects by a possible strategy that is the optimization of the sgRNA. This is not the only strategy that has been proposed. At least another strategy is by protein mutations to modify molecular interactions. Thus, the authors should include in the introduction a comment on the fact that their approach is a possibility, but not the only possibility, and give credit to other strategies by citing appropriate references, both experimental and computational.”

In response to this comment, the authors have added a short sentence and a reference on a design strategy based on protein engineering. The added reference concerns protein mutations that should change the **hydrogen bond pattern between protein and DNA**.

There is another strategy for protein engineering, which targets **electrostatic interactions between protein and DNA**. I think it is also fair to cite that possibility, but I totally leave it to the authors' choice.

- Slaymaker et al., *Science* **2016**, 351, 84.
- Ray et al., *J. Phys. Chem. B* **2020**, 124, 2168.

Reviewer #3:

Remarks to the Author:

The authors did a great job in improving their manuscript. Some of my comments have been successfully addressed, and the manuscript has indeed been improved. However, there are still some points that need to be addressed for publication.

Regarding my comment (minor comment 4) "To help readers reproduce the results described in the manuscript, the authors should summarize all sgRNA sequences and target sequences from Guide-seq, Change-seq, and Circle-seq used in the machine learning process, along with the energy calculation values, key feature calculation values, all feature values utilized in the development of the CRISOT-FP machine learning model, and model prediction values, and present this information in a table.", the authors have not yet provided the data used for their machine learning such as sgRNA and target sequences, calculated feature values for each pair of sgRNA and target sequences (e.g. resA@P resA@O5'). This information is essential for the reproduction of authors' conclusions. If the authors performed machine learning, they should have organized training data directly used for their machine learning the training data readily available, which was directly used for their machine learning.

Regarding my comment (minor comment 11) "In Fig. 5a, only four sgRNAs were tested for validation purposes, which is an insufficient number. It is recommended to test at least 30 sgRNAs for validation. If whole genome sequencing is too expensive, it is suggested to use more commonly used methods such as Guide-seq, Change-seq, or Circle-seq, which are described in other papers, for validation experiments.", the authors have declined to conduct additional experiments. They stated that their study is limited in that their conclusions were not extensively validated by wet experiments. As a result, the paper would be better suited for a more specialized journal, such as Communication Biology.

Regarding my comment (minor comment 5) "In Fig. 2f, g, h, i, j, and k, 11 computer models were compared; however, in some figures, only 8 comparison bars are shown. Please ensure that all 11 models are shown in all relevant figures. Additionally, in Fig. 3e and 3f, Fig. 4d and 4e, and Fig. 5c, please include a comparison of all 11 computer models rather than comparing only some of them.", the authors acknowledged that they did not present certain results because the training datasets partially overlapped with the test datasets. Then, the authors could generate new test data by excluding sgRNAs used for training and display the results using hatched colors to indicate the modified test data.

Regarding the sentence "The atom position features represented by distances, angles and dihedral angles of atoms were calculated using pytraj package.", more information is necessary about the "pytraj package"; the authors may add references to provide more details about this package.

Regarding comments from reviewer 2, the authors also did a great job addressing most of them, thereby improving the manuscript. However, some comments still require more thorough responses.

1. Regarding "In the comparison to other methods (Fig 2f-k) have the authors made sure that none of the methods compared have been trained on the data used for testing? For example comparing on guide-seq is that a subset that was not used for training some of the methods? This is obviously critical in making a fair comparison. The authors should clarify this and provide sets of gRNAs in supplementary material on precisely which off-target for which gRNAs the methods are compared on and flag the subsets used in the respective trainings.", the authors now provide only

binary information regarding whether the test data contain sgRNAs that were used for training. However, quantitative descriptions would be clearer when comparing the performance of the models. The authors can provide the number of sgRNAs used for training out of the total number in the test data (e.g., 3/5,000). In addition, reviewer 2 recommended providing the sets of sgRNAs in supplementary materials but the authors have not address this point. I would recommend presenting the training and test datasets in spreadsheets that detail the target sequences, sgRNA sequences, values of model features, and indicate whether the data were used for training. This transparent supplementary information would be essential for reproducing authors' conclusions.

2. Regarding the following comment and authors' reply

"Since the authors use molecular dynamics, I wonder why the authors are not comparing to kinetics base methods such as Eslami-Mossallam et al., NCOMM, 2022 and Fu et al., NCOMM, 2022. The latter also considers sequence determinants for CRISPR specificity which seems to be highly relevant to include in the authors study. The authors should take these methods into account in their work.

Authors' reply: Following the reviewer's comment, we have adopted the MOFF (Eslami-Mossallam et al., NCOMM, 2022) and KinPred (Fu et al., NCOMM, 2022) off-target prediction methods as comparison methods in the validation studies."

I have found that these two methods are missing from the list of compared methods (e.g., Fig. 3e).

We would like to appreciate the reviewers for their thoughtful comments and insightful suggestions to improve our manuscript. We genuinely enjoyed incorporating the reviewers' comments, and the manuscript has been carefully revised (colored in **red**) in response to the comments.

Summary of the major changes:

(1) We have **reorganized the Group I-III datasets**. The Group I datasets were used for training, whereas the Group II-III datasets, which are independent without overlaps to the Group I datasets, were used for testing. **Systematic benchmark validations** were thus presented for CRISOT-FP, CRISOT-Score and CRISOT-Spec respectively, among them, three benchmark tests were performed on the Group II datasets for CRISOT-FP, and an additional benchmark validation were performed on the Group III datasets for CRISOT-FP, CRISOT-Score and CRISOT-Spec. The manuscript is revised accordingly.

(2) We have **clearly clarified the difference between the training and testing datasets**, and presented the results intuitively using **gray colors and slash hatches**. For the evaluation of CRISOT-FP, two of the compared learning-based methods were trained in their original studies on datasets containing several sgRNAs that were contained in our testing datasets. The results are summarized in **Fig. R1**. Therefore, in the case that all testing sgRNAs were contained in the training datasets of the compared methods, we have **indicated this case by using the bars in gray**. For example, although the performances of DLcrispr and CRISPRnet were higher/comparable to those of CRISOT-FP in the in vitro datasets (**Fig. R1a**), this is attributed to that the testing sgRNAs are already contained in their training datasets, thus the colors of their bars were changed to gray to indicate potential data leakage. In the case that the testing sgRNAs partially overlapped with the training datasets of the compared methods, we have **presented additional bars in slash hatch**. For example, data leakage may exist in **Fig. R1b** and **Fig. R1d** because the training datasets of the DLcrispr and CRISPRnet models contained sgRNAs that were contained in the testing datasets. Therefore, we removed the sgRNAs in the testing datasets that were contained in the training datasets of DLcrispr and CRISPRnet, and presented the results with slash hatch. For a fair comparison, the same sgRNAs were removed in the testing datasets for CRISOT-FP, and the results were also presented with slash hatch. It can be seen that the performances of CRISOT-FP and DLcrispr were comparable without excluding the overlaps, while CRISOT-FP significantly surpassed DLcrispr after removing the overlaps (**Fig. R1d**). Taken together, these results clearly indicated that CRISOT-FP, which serves as the basis of the following modules CRISOT-Score, CRISOT-Spec and CRISOT-Opti, significantly outperformed the compared methods without potential data leakage.

Fig. R1 Summary of the benchmark validation results on the Group II and III datasets. Averaged PR-AUC results for (a) *in vitro* datasets, (b) in cell datasets, (c) targeted (Surro-seq) dataset with different cutoff of fold-change (FC) and (d) prime/base editor off-target datasets are shown. The blue and green bars represent learning-based and hypothesis-driven methods, respectively. Bars in gray color are those the training datasets contained all sgRNAs in the testing datasets. Bars with slash hatch are presented by excluding the overlapped sgRNAs. Of note, for fair comparisons, the same sgRNAs were removed from the testing datasets for CRISOT-FP. The overlapped sgRNAs are summarized in **Supplementary Data 4**. The CRISOT-FP models are trained on the Group I datasets, which are independent to the testing datasets. Paired t-tests were performed. n.s.: not significant, *: $p < 0.05$, **: $p < 0.01$, ***: $p < 0.001$.

(3) We have carefully **reorganized the required data**, including the features, the SHAP values and the training/testing datasets, and provided them as the **Supplementary Data 1~3**. We have also improved the off-target analysis by presenting the detailed sgRNA overlapping information in **Supplementary Data 4**. All required supplementary data have been uploaded to the manuscript submission system.

Our point-to-point responses can be found below. For clarity, the reviewers' comments are colored in **black** and our responses are in **blue**.

Reviewer #1 (Remarks to the Author):

The authors have exhaustively addressed all my comments and I recommend the current version of the manuscript for publication in Nature Communications. I just wish to add an optional recommendation. It is related to my previous comment. "The authors address the issue of off-target effects by a possible strategy that is the optimization of the sgRNA. This is not the only strategy that has been proposed. At least another strategy is by protein mutations to modify molecular interactions. Thus, the authors should include in the introduction a comment on the

fact that their approach is a possibility, but not the only possibility, and give credit to other strategies by citing appropriate references, both experimental and computational.”

In response to this comment, the authors have added a short sentence and a reference on a design strategy based on protein engineering. The added reference concerns protein mutations that should change the hydrogen bond pattern between protein and DNA.

There is another strategy for protein engineering, which targets electrostatic interactions between protein and DNA. I think it is also fair to cite that possibility, but I totally leave it to the authors’ choice.

- Slaymaker et al., Science 2016, 351, 84.
- Ray et al., J. Phys. Chem. B 2020, 124, 2168.

Authors’ reply: We thank the reviewer for the comment. We have added the two citation for protein engineering strategy.

Reviewer #3 (Remarks to the Author):

The authors did a great job in improving their manuscript. Some of my comments have been successfully addressed, and the manuscript has indeed been improved. However, there are still some points that need to be addressed for publication.

Regarding my comment (minor comment 4) “To help readers reproduce the results described in the manuscript, the authors should summarize all sgRNA sequences and target sequences from Guide-seq, Change-seq, and Circle-seq used in the machine learning process, along with the energy calculation values, key feature calculation values, all feature values utilized in the development of the CRISOT-FP machine learning model, and model prediction values, and present this information in a table.”, the authors have not yet provided the data used for their machine learning such as sgRNA and target sequences, calculated feature values for each pair of sgRNA and target sequences (e.g. resA@P resA@O5’). This information is essential for the reproduction of authors’ conclusions. If the authors performed machine learning, they should have organized training data directly used for their machine learning the training data readily available, which was directly used for their machine learning.

Authors’ reply: Thanks for this comment.

About the training/testing datasets: We had provided all datasets on the CRISOT github website (<https://github.com/bm2-lab/CRISOT>) in the first revision, since the size of the dataset was too big. In response to this comment, we have packaged the datasets and uploaded it as the **Supplementary Data 3**.

About the features: The features used for encoding the sgRNA-DNA pairs were given and described in the former **Table S1**. The reviewer commented to provide the encoded features for each sgRNA-DNA pair. However, the feature dimension ($193 * 20 = 3860$) is so large that it is impossible to upload such a huge file with millions of sgRNA-DNA pairs. Therefore, we have developed a python script for encoding a given pair of sgRNA-DNA sequences in the source code of CRISOT (<https://github.com/bm2-lab/CRISOT>). We have added 3 sgRNA-DNA pairs as example in the former **Table S1** and packaged it as the **Supplementary Data 1**.

We have also reorganized the feature importance values and SHAP values in the former **Tables S2** and **S3**, and packaged them as the **Supplementary Data 2**.

Now all required data are available as **Supplementary Data 1-3**.

Regarding my comment (minor comment 11) “In Fig. 5a, only four sgRNAs were tested for validation purposes, which is an insufficient number. It is recommended to test at least 30 sgRNAs for validation. If whole genome sequencing is too expensive, it is suggested to use more commonly used methods such as Guide-seq, Change-seq, or Circle-seq, which are described in other papers, for validation experiments.”, the authors have declined to conduct additional experiments. They stated that their study is limited in that their conclusions were not extensively validated by wet experiments. As a result, the paper would be better suited for a more specialized journal, such as Communication Biology.

Authors’ reply: Thanks. We agreed with the reviewer that extensive validation is a good supplement. However, as a methodology paper, the main contributions of our study is to develop a computational

package CRISOT for sgRNA design based on AI and MD technologies, and the performance of this tool is evaluated extensively. We would like to clarify this point further in the following:

1) The work for **Fig. 5a (now Fig. 6a)** lies in the optimization of sgRNA, *i.e.* the CRISOT-Opti module, which is the application extension of CRISOT-FP, CRISOT-Score and CRISOT-Spec. In the current study we just focus on presenting such novel optimization strategy as a proof of concept study by using MD simulations and AI technology, and its effectiveness is already experimentally validated, although limited, with 2+4 sgRNAs on three important genes using Guide-seq and WGS assays. In addition, its effectiveness is guaranteed by the powerful and state-of-the-art prediction performance of the former CRISOT modules that have been systematically validated using the Group II and Group III datasets.

2) We have performed two kinds of off-target detection assays, namely the Guide-seq and WGS assays, on 2+4 sgRNAs for three important genes. The Guide-seq and WGS assays are valid genome-wide off-target detection method with different advantages. Performing different experiments helped to improve the generalization and make the results more validated, even though the number of tested sgRNAs was limited.

3) We think whether additional experiments are necessarily needed depends on whether our computational benchmark validations are solid. We now have clearly presented and clarified the benchmark validations of CRISOT-FP, which serves as the basis of the CRISOT-Opti and the other CRISOT modules, and we now clearly indicated that the testing datasets are completely non-overlapping with the training datasets. The manuscript is revised accordingly.

In the case that all testing sgRNAs were contained in their training datasets of the compared methods, we have indicated this case by using the bars in gray. For example, although the performances of DLcrispr and CRISPRnet were higher/comparable to those of CRISOT-FP in the *in vitro* datasets (**Fig. R1a** and **Fig. 3a**), this is attributed to that the testing sgRNAs are already contained in their training datasets, thus the colors of their bars were changed to gray to indicate potential data leakage.

In the case that the testing sgRNAs partially overlapped with the training datasets of the compared methods, we have presented additional bars in slash hatch. For example, data leakage may exist in **Fig. R1b**, **Fig. R1d**, **Fig. 3b** and **Fig. 6b** because the training datasets of the DLcrispr and CRISPRnet models contained sgRNAs that were contained in the testing datasets. Therefore, we removed the sgRNAs in the testing datasets that were contained in their training datasets of DLcrispr and CRISPRnet, and presented the results with slash hatch. For a fair comparison, the same sgRNAs were removed in the testing datasets for CRISOT-FP, and the results were also presented with slash hatch. It can be seen that the performances of CRISOT-FP and DLcrispr were comparable without excluding the overlaps, while CRISOT-FP significantly surpassed DLcrispr after removing the overlaps (**Fig. R1d** and **Fig. 6b**).

Taken together, these results clearly indicated that CRISOT-FP significantly outperformed the compared methods without potential data leakage.

Fig. 3 Comparison of CRISOT-FP with the state-of-the-art off-target prediction methods in predicting independent (the Group II) datasets. PR-AUC results for (a) *in vitro* datasets, (b) in cell datasets and (c) targeted (Surro-seq) dataset with different cutoff of fold-change (FC) are shown, and the ROC-AUC results can be found in Fig. S6-11. The blue and green bars represent learning-based and hypothesis-driven methods, respectively. Bars in gray color are those the training datasets contained all sgRNAs in the testing datasets. Bars with slash hatch are presented by excluding the overlapped sgRNAs. Of note, for fair comparisons, the same sgRNAs were removed from the testing datasets for CRISOT-FP. The overlapped sgRNAs are summarized in **Supplementary Data 4**. The CRISOT-FP models are trained on the Group I datasets, which are independent to the Group II datasets.

Fig. 6b Comparison of CRISOT-FP and CRISOT-Score with the existing off-target prediction methods for predicting the off-target effects of base and prime editors using PR-AUC. Learning-based methods are colored in blue, hypothesis-driven methods are colored in green. Bars in gray color are those the training datasets contained all sgRNAs in the testing datasets. Bars with slash hatch are presented by excluding the overlapped sgRNAs. Of note, for fair comparisons, the same sgRNAs were removed from the testing datasets for CRISOT-FP. The overlapped sgRNAs are summarized in **Supplementary Data 4**. The CRISOT-FP models are trained on the Group I datasets, which are independent to the Group III datasets.

Taking together, the CRISOT-FP, which serves as the basis of the CRISOT-Opti and the other CRISOT modules, is systematically validated, the results are clearly presented and clarified, and its effectiveness and generalizability are guaranteed by the independent Group II and Group III datasets. In this case, we agreed with the reviewer that extensive validation for CRISOT-Opti is a good supplement, while we think as a proof-of-concept study, the effectiveness of CRISOT-Opti is demonstrated clearly. The large scale experimental validation for the optimization of sgRNA is out of the scope of our current study, and it is better to be performed separately as an independent work in the future.

Regarding my comment (minor comment 5) “In Fig. 2f, g, h, i, j, and k, 11 computer models were compared; however, in some figures, only 8 comparison bars are shown. Please ensure that all 11 models are shown in all relevant figures. Additionally, in Fig. 3e and 3f, Fig. 4d and 4e, and Fig. 5c, please include a comparison of all 11 computer models rather than comparing only some of them.”, the authors acknowledged that they did not present certain results because the training datasets partially overlapped with the test datasets. Then, the authors could generate new test data by excluding sgRNAs used for training and display the results using hatched colors to indicate the modified test data.

Authors’ reply: We thank the reviewer for this comment.

We have summarized the overlapping information in **Supplementary Data 4**. The sgRNAs of the training datasets of the compared models CRISPRnet and DLcrispr in their original studies totally overlapped datasets including the Circle-seq and Guide-seq_tsai datasets, and partially overlapped datasets including the Guide-seq_listgarten and PE2_Kim datasets. Following the reviewer's comment, we have presented both the results on the complete testing datasets and the results without overlapped sgRNAs for these models (**Fig. 3**, **Fig. 6b**, **Fig. S6-S11** and **Fig.S26-S27**), and presented the results intuitively using gray colors and slash hatches for totally and partially overlapped methods, respectively. In the case that all testing sgRNAs were contained in their training datasets of the compared methods, we have indicated this case by using the bars in gray. In the case that the testing sgRNAs partially overlapped with the training datasets of the compared methods, we have presented additional bars in slash hatch. Detailed descriptions and examples can be found in the response to the 2nd comment of

Reviewer #3. The results showed that CRISOT-FP significantly ($p < 0.05$) outperformed the compared methods (**Fig. R1** attached to this letter).

Regarding the sentence “The atom position features represented by distances, angles and dihedral angles of atoms were calculated using pytraj package.”, more information is necessary about the “pytraj package”; the authors may add references to provide more details about this package.

Authors’ reply: Thanks for this comment. The pytraj package is a Python front-end of CPPTRAJ¹ from AmberTools. We have added a short description and a reference in the Methods section.

Regarding comments from reviewer 2, the authors also did a great job addressing most of them, thereby improving the manuscript. However, some comments still require more thorough responses.

1. Regarding “In the comparison to other methods (Fig 2f-k) have the authors made sure that none of the methods compared have been trained on the data used for testing? For example comparing on guide-seq is that a subset that was not used for training some of the methods? This is obviously critical in making a fair comparison. The authors should clarify this and provide sets of gRNAs in supplementary material on precisely which off-target for which gRNAs the methods are compared on and flag the subsets used in the respective trainings.”, the authors now provide only binary information regarding whether the test data contain sgRNAs that were used for training. However, quantitative descriptions would be clearer when comparing the performance of the models. The authors can provide the number of sgRNAs used for training out of the total number in the test data (e.g., 3/5,000). In addition, reviewer 2 recommended providing the sets of sgRNAs in supplementary materials but the authors have not address this point. I would recommend presenting the training and test datasets in spreadsheets that detail the target sequences, sgRNA sequences, values of model features, and indicate whether the data were used for training. This transparent supplementary information would be essential for reproducing authors’ conclusions.

Authors’ reply: We thank the reviewer for this comment.

In the first part of this comment, the reviewer suggested to quantitatively summarize the overlapped sgRNAs. As is mentioned in the response to the 3rd comment of Reviewer #3, we have summarized the overlapping information in **Supplementary Data 4**. We have also presented both the results on the complete testing datasets and the results without overlapped sgRNAs for these models (**Fig. 3, Fig. 6b, Fig. S6-S11** and **Fig.S26-S27**). The results showed that CRISOT-FP significantly ($p < 0.05$) outperformed the compared methods (**Fig. R1** attached to this letter).

In the second part of this comment, the reviewer suggested to provide the detailed datasets. As mentioned in the response to the first comment of Reviewer #3, we have uploaded the detailed datasets as **Supplementary Data 3**, and the detailed descriptions are given in the Methods section. Models are trained on Group I datasets, and tested on Groups II and III datasets. We do not provide the encoded features for each sgRNA-DNA pair because the total size is huge. Instead, we have provided the raw features and several instances as encoding examples in **Supplementary Data 1**, and provided a python script for the readers to encode their own inputs.

2. Regarding the following comment and authors’ reply

“Since the authors use molecular dynamics, I wonder why the authors are not comparing to kinetics base methods such as Eslami-Mossallam et al., NCOMM, 2022 and Fu et al., NCOMM, 2022. The latter also considers sequence determinants for CRISPR specificity which seems to be highly relevant to include in the authors study. The authors should take these methods into account in their work.

Authors’ reply: Following the reviewer’s comment, we have adopted the MOFF (Eslami-Mossallam et al., NCOMM, 2022) and KinPred (Fu et al., NCOMM, 2022) off-target prediction methods as comparison methods in the validation studies.”,

I have found that these two methods are missing from the list of compared methods (e.g., Fig. 3e)

Authors’ reply: Thanks for this comment. The **Fig. 3e** (now **Fig. 4e**) compared CRISOT-Score, the off-target scoring method developed using the key interaction features, to the state-of-the-art off-target scoring methods. Therefore, learning-based methods, including the MOFF model, were not included in **Fig. 4e**. The comparison of CRISOT with all learning-based methods, including the MOFF model, can be found in **Fig. 3**. As for KinPred, which is a scoring method, its results were presented as the second

bar in each group in **Fig. 4e**. These results indicated that CRISOT-FP significantly outperformed MOFF, and CRISOT-Score significantly outperformed KinPred in all benchmark scenarios. In response to this comment, we have added a short comment in bracket to indicate that "learning-based methods were not included" in this comparison in Section 3.2.

Fig. 4e Comparison of CRISOT-Score with the state-of-the-art off-target scoring methods. The PR-AUC results indicate that CRISOT-Score surpasses the existing off-target scoring methods. The results of KinPred are presented as the second bar in each group.

References

1. Roe, D. R. & Cheatham, T. E. PTRAJ and CPPTRAJ: Software for processing and analysis of molecular dynamics trajectory data. *J. Chem. Theory Comput.* **9**, 3084–3095 (2013).

Reviewers' Comments:

Reviewer #3:

Remarks to the Author:

The authors' 2nd revision has enhanced the quality of the manuscript.

I have a single outstanding concern. The authors once again declined to conduct the additional validation experiments I requested in the initial review, citing their belief that their research serves as a proof-of-concept study. Nevertheless, I believe that without the suggested validation studies, the evidence underpinning the models' performance is not sufficiently convincing. The final decision regarding the paper's suitability for publication, of course, rests with the editor.

We would like to appreciate reviewer #3 for his/her comments to improve our manuscript. Our point-to-point responses can be found below. For clarity, the reviewer's comments are colored in **black** and our responses are in **blue**.

Reviewer #3 (Remarks to the Author):

The authors' 2nd revision has enhanced the quality of the manuscript.

I have a single outstanding concern. The authors once again declined to conduct the additional validation experiments I requested in the initial review, citing their belief that their research serves as a proof-of-concept study. Nevertheless, I believe that without the suggested validation studies, the evidence underpinning the models' performance is not sufficiently convincing. The final decision regarding the paper's suitability for publication, of course, rests with the editor.

Authors' reply: We thank the reviewer for this comment.

We agreed with the reviewer that extensive experimental validation study is a good supplement to this study. Although extensive in-silico validations are performed on CRISOT, we tone down the advances of this work with respect to the experimental validation (highlighted in the manuscript). The words new/novel/first and 'extremely'/'outstanding' have been checked/revised. We also expect that large-scale experimental validations can be performed in the future.